# Sleep drive, not total sleep amount, increases seizure risk

**Vishnu Anand Cuddapah** [1,2] ✉, **Cynthia T. Hsu**[3,4,7], **Fernanda Valle Sirias**[1,2], **Yongjun Li** [3,4], **Hrishit M. Shah**[3,4], **Christopher Saul**[3,4,8], **Samantha Killiany**[3,4], **Camilo Guevara** [3,4], **Joy Shon**[3,4], **Zhifeng Yue**[3,4], **Gabrielle L. Gionet**[5,6], **Mary E. Putt** [5,6] **& Amita Sehgal** [3,4] ✉

Sleep loss has been associated with increased seizure risk since antiquity. Using automated video detection of spontaneous seizures in *Drosophila* epilepsy models, we show that seizures worsen only when sleep restriction raises homeostatic "sleep drive," not simply when total sleep amount falls. This is supported by the paradoxical finding that acute activation of sleep-promoting circuits worsens seizures, because it increases sleep drive without changing sleep amount. Sleep-promoting circuits become hyperactive after sleep loss and are associated with increased whole-brain activity. During sleep restriction, optogenetic inhibition of sleep-promoting circuits to reduce sleep drive protects against seizures. Downregulation of the 5HT1A serotonin receptor in sleep-promoting cells mediates the effect of sleep drive on seizures, and we identify an FDA-approved 5HT1A agonist to mitigate seizures. Our findings demonstrate that while homeostatic sleep is needed to recoup lost sleep, sleep drive comes at the cost of increasing seizure susceptibility.

Since the writings of Hippocrates and Aristotle into the modern era, poor sleep has been associated with increased seizure risk[1]. People with epilepsy, as well as caregivers of children with epilepsy, report that sleep disruption is a common trigger for seizure exacerbation[2,3]. Understanding how sleep restriction might increase seizure risk has been challenging because of multiple confounding variables that change through the day, including light exposure, food intake, sleep and wake, physiological processes regulated by endogenous circadian clocks, etc. However, when well-defined protocols are used, distinct contributions of sleep on brain excitability have been suggested[4,5] although the mechanisms that drive this are not understood.

To understand how sleep restriction increases seizure severity, we leveraged the tractability of *Drosophila*. Sleep loss leads to diffuse and focal effects in the *Drosophila* brain. On a broad level, synapse number and size increase in the fly brain with increased time awake[6]. At a circuit/cellular level, sleep deprivation leads to increased activity of the dorsal fan-shaped body (dFB)[7,8] and ellipsoid body[9], which drives increased sleep. Specific Kenyon cells in the mushroom body (MB)[10–12], subsets of neurons in the pars intercerebralis (PI)[13,14], and 2 cholinergic neurons in the ventral nerve cord (VNC)[15,16] also have sleep-promoting roles. In addition, several loci in the fly brain drive wakefulness[8,12,17]. In the present study, we investigated if activity of sleep:wake-regulating circuits modulates seizure risk.

[1]Jan and Dan Duncan Neurological Research Institute, Texas Children's Hospital, Houston, TX, USA. [2]Division of Pediatric Neurology and Developmental Neuroscience, Department of Pediatrics, Baylor College of Medicine, Houston, TX, USA. [3]Howard Hughes Medical Institute, University of Pennsylvania, Philadelphia, PA, USA. [4]Chronobiology and Sleep Institute, Perelman School of Medicine, University of Pennsylvania, Philadelphia, PA, USA. [5]Department of Biostatistics, Epidemiology and Informatics University of Pennsylvania, Philadelphia, PA, USA. [6]CHOP/Penn Intellectual and Developmental Disabilities Research Center, Philadelphia, PA, USA. [7]Present address: Department of Biology, California State University Fresno, Fresno, CA, USA. [8]Present address: Department of Chemical & Biomolecular Engineering, University of Delaware, Newark, DE, USA. ✉e-mail: Vishnu.Cuddapah@bcm.edu; amita@pennmedicine.upenn.edu

## Results

### Sleep restriction worsens induced seizures

To first understand if sleep loss is associated with seizure severity in *Drosophila*, we took advantage of bang-sensitive neurogenetic models of epilepsy. Bang-sensitive flies are prone to having seizures after sustaining a strong mechanical stimulus, e.g. after flies are banged against a countertop or vortexed. Like mammalian models, seizures in flies occur as a sequence of stereotyped behaviors: 1) an atonic "paralytic" phase, 2) a tonic/clonic "convulsive" phase, and a 3) post-ictal recovery phase (Fig. 1a)[18]. While most studies monitor the total time to recovery from seizures, we sought to assess seizure severity by timing how long each fly spent in each of these discrete phases; this was achieved through video recording followed by time measurements (Fig. 1a and Supplementary Movie 1).

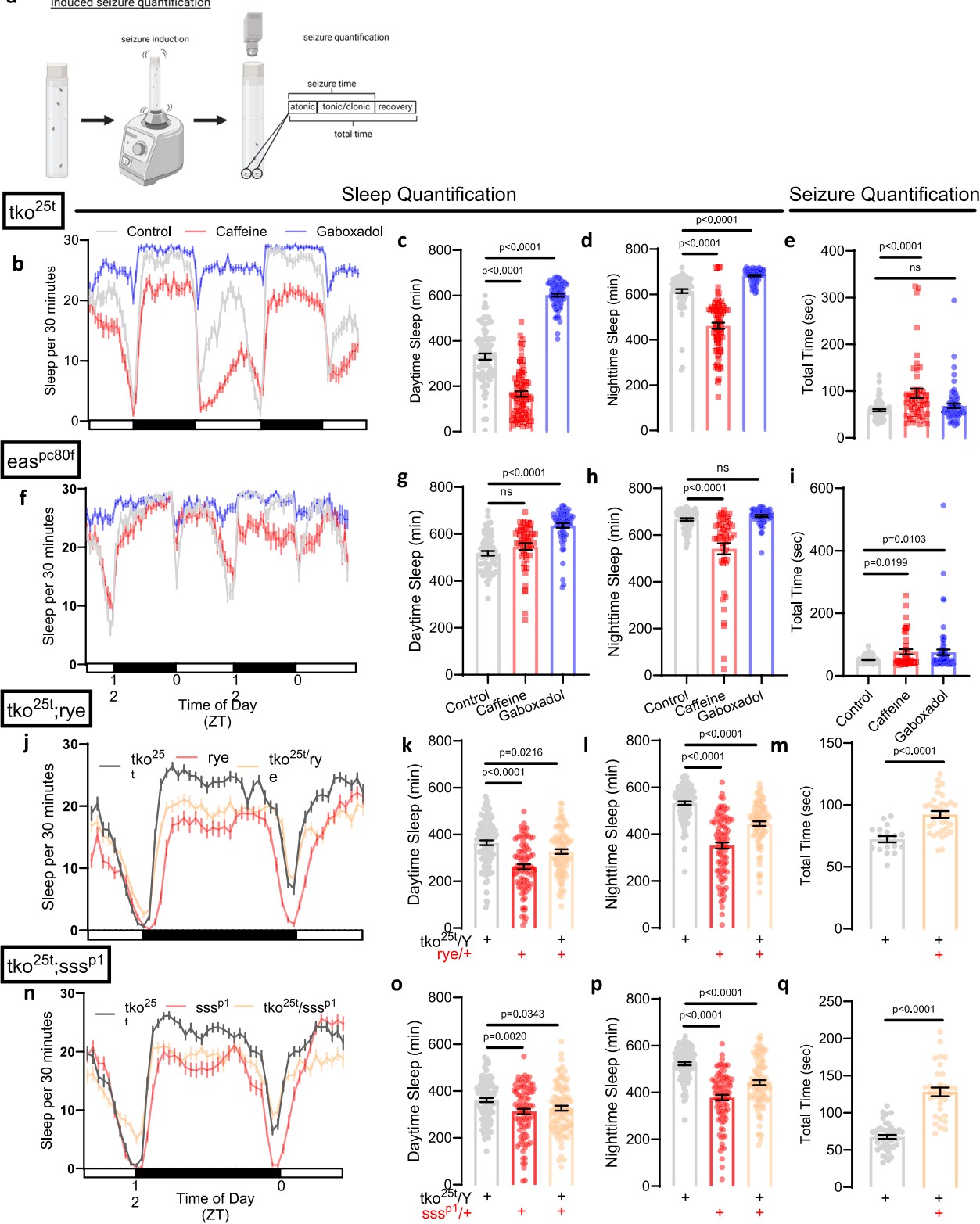

**Fig. 1 | Sleep loss leads to more severe induced seizures in bang-sensitive _tko²⁵ᵗ_ and _eas^{pc80f}_ mutant flies. a** Experimental protocol depicting seizure induction on vortexer followed by video recording of fly seizures. Seizures are quantified for atonic ("paralysis"), tonic/clonic ("convulsive"), and recovery ("postictal") phases. "Seizure time" is atonic + tonic/clonic phases. "Total time" is "seizure time" + recovery phase. Created in BioRender. Sehgal, A. (2025) https://BioRender.com/p4jrytx. **b–d** _tko²⁵ᵗ_ flies exhibit decreased mean sleep times with caffeine and increased mean sleep times with gaboxadol. n = 78 flies/condition. **e** _tko²⁵ᵗ_ flies demonstrate prolonged mean seizure duration with caffeine treatment (n = 51 flies) as compared to control (n = 74 flies). No significant effect was seen with gaboxadol treatment (n = 67 flies). **f–h** _eas^{pc80f}_ flies exhibit decreased mean nighttime sleep times with caffeine (n = 47 flies) and increased mean sleep times with gaboxadol (n = 62 flies) as compared to control (n = 59 flies). **i** _eas^{pc80f}_ flies demonstrate prolonged mean seizure times with caffeine (n = 43 flies) and gaboxadol (n = 71 flies) as compared to control (n = 88 flies). **j–l** _tko²⁵ᵗ;rye_ (n = 96 flies) exhibit decreased mean daytime and nighttime sleep times as compared to _tko²⁵ᵗ_ and _rye_ flies (n = 95 flies/condition). **m** _tko²⁵ᵗ;rye_ flies (n = 33 flies) demonstrate prolonged mean seizure times as compared to _tko²⁵ᵗ_ flies (n = 18 flies). **n–p** _tko²⁵ᵗ;sss^{p1}_ flies (n = 94 flies) and _sss^{p1}_ flies (n = 94 flies) exhibit decreased mean daytime and nighttime sleep times as compared to _tko²⁵ᵗ_ flies (n = 96 flies). **q** _tko²⁵ᵗ;sss^{p1}_ flies (n = 33 flies) demonstrate prolonged mean seizure times as compared to _tko²⁵ᵗ_ flies (n = 44 flies). Two-group two-tailed t-test or one-way ANOVA with Dunnett's or Tukey's multiple comparisons test was used. Data are presented as mean values ± SEM. Source data are provided as a Source Data file.

Given that each method of sleep restriction is associated with different confounding variables, we restricted sleep using multiple approaches. We reasoned that a consistent effect across distinct modalities of sleep restriction would truly reflect a function of sleep loss. We first sleep restricted _tko²⁵ᵗ_ mutant flies, which exhibit bang-sensitive seizures, through caffeine feeding as previously described[19]. Caffeine led to sleep loss, and this was associated with prolonged seizures and recovery times in _tko²⁵ᵗ_ flies (Fig. 1b–e, Supplementary Fig. 1a–d). We also assessed _eas^{pc80f}_ bang-sensitive flies, which demonstrated a loss of nighttime sleep after caffeine exposure and an increase in total seizure time (Fig. 1f–i, Supplementary Fig 1e–h). In the absence of caffeine-induced sleep loss, _eas^{pc80f}_ mutant flies did not exhibit a tonic/clonic phase; this phase only appears after sleep restriction (Supplementary Fig. 1n). Caffeine treatment led to increased fly death in bang-sensitive mutants but not wild-type flies (Supplementary Fig. 1j, l, o), correlating with increased seizure severity. These data demonstrate that sleep restriction worsens seizures in flies consistent with previous findings[20].

Caffeine may have off-target effects, so we sought to use additional methods of sleep restriction to understand the effects on seizure severity. We attempted to use physical manipulation to restrict sleep, but this resulted in bang-sensitive seizures during the protocol, pushing flies into a subsequent refractory state resistant to seizure induction. We then turned to genetic manipulations to restrict sleep. _tko²⁵ᵗ_ mutant flies were crossed to _redeye_ mutant flies (_rye_) or _sleepless_ mutant flies (_sss^{p1}_), which are short-sleeping mutant flies exhibiting a significant reduction in sleep amount. _tko²⁵ᵗ_ flies crossed to these short-sleeping mutants exhibited a decrease in sleep with prolongation of seizure times, mostly driven by a longer tonic-clonic phase (Fig. 1j–q; Supplementary Fig. 2a–j). Taken together, these data demonstrate that both pharmacologically- and genetically-induced sleep loss leads to worsened seizures.

Given that sleep loss is associated with seizure exacerbation, we hypothesized that sleep enhancement would protect against seizures. To test this hypothesis, we fed bang-sensitive flies gaboxadol (also known as THIP), an extrasynaptic, δ-subunit-containing GABA_A receptor agonist[21] known to promote sleep[22], and measured seizure severity. Gaboxadol treatment reliably increased sleep amount across genotypes (Fig. 1b–d, f–h); surprisingly, it was not protective against seizures and even prolonged seizures in _eas^{pc80f}_ mutant flies (Fig. 1i). Given possible sleep-independent effects of gaboxadol, we also sought to enhance sleep using a more specific genetic approach. Overexpression of UAS-_nemuri_ using an inducible, pan-neuronal Gal4 driver (_elav_-GeneSwitch, _elav_-GS) leads to increased sleep[23]. Induction of neuronal UAS-_nemuri_ expression in the _tko²⁵ᵗ_ background by feeding flies the GeneSwitch activator RU486 led to an increase in sleep (Supplementary Fig. 3a–c). This sleep induction led to a complete block of seizures (Supplementary Fig. 3d). A smaller sleep induction was also noted in the absence of RU486 (Supplementary Fig. 3b), likely indicating leaky _nemuri_ expression. Even this small sleep induction was sufficient to decrease seizure times in _tko²⁵ᵗ_ flies (Supplementary Fig. 3e–i). To determine if the protection against seizures was due to a direct effect of _nemuri_ or an effect of increased sleep, _tko²⁵ᵗ_ flies were crossed with _nemuri_ mutant flies (_nur³_). Sleep was similar in _tko²⁵ᵗ_ and _tko²⁵ᵗ/nur³_ flies (Supplementary Fig. 4a–c), consistent with previous findings that loss of _nemuri_ does not change daily sleep amount[23]. Loss of _nemuri_ did not have a significant effect on seizures (Supplementary Fig. 4d–i). This is consistent with the conclusion that sleep enhancement, and not direct effects of _nemuri_, is responsible for seizure protection. Across the bang-sensitive mutants, sleep enhancement with gaboxadol had no effect or worsened seizures, but sleep enhancement with _nemuri_ blocked seizures. This indicated that other variables related to sleep beyond purely sleep duration might be important drivers of seizure severity.

## Development of a video-tracking platform to identify spontaneous seizures in flies

Unexpectedly, while caffeine-induced sleep loss increased seizure severity, it decreased the likelihood of subsequent induced seizures in _eas^{pc80f}_ mutant flies (Supplementary Fig. 1m), similar to previously noted effects of mechanical sleep restriction on temperature-sensitive seizures[24]. Bang-sensitive seizures are known to be less likely if flies are in a refractory state due to recent seizures[25,26]; this raised the possibility that undocumented spontaneous seizures were occurring prior to the bang-sensitive assay while the flies were being sleep-deprived with caffeine. The occurrence of spontaneous seizures has been suspected in fly models of epilepsy[27], but the lack of non-invasive electroencephalography has made this challenging to confirm. Therefore, we sought to create a method to identify spontaneous seizures in flies.

We developed a chronic video-tracking pipeline to automatically detect and quantify spontaneous, previously undocumented, seizures in flies (Fig. 2a). In this paradigm, flies were placed into individual wells of 24- or 48-well plates and video recorded for 96 hours in 12-hour light:12-hour dark cycles using an infrared camera at 25 frames per second. Video tracking software was used to convert fly body positions to XY coordinates over time. We then identified velocity, acceleration, and pixel change thresholds that are typical of convulsive seizures and never occur in Canton-S or _w1118_ wild-type flies (CynthiSeize algorithm; see Supplementary Methods). This pipeline not only enabled robust identification of spontaneous (non-induced) seizures but also simultaneous quantification of sleep state, sleep amount, and seizure lethality for individual flies (see Supplementary Movie 2, 3, and 4 for samples of spontaneous seizures). To further validate that the identified hyperkinetic tonic-clonic behavioral sequences were indeed seizures, we pan-neuronally expressed TRIC-luciferase, a GFP-based calcium reporter coupled to luciferase that allows for measurement of neural activity in awake behaving flies[28]. This demonstrated that the hyperkinetic tonic-clonic behavioral sequence in flies was associated with prolonged hypersynchronous neuronal activation (Supplementary Fig. 5), consistent with seizures. In addition, seizures were ameliorated with well-established anti-seizure medications including levetiracetam and sodium valproate (Supplementary Fig. 6).

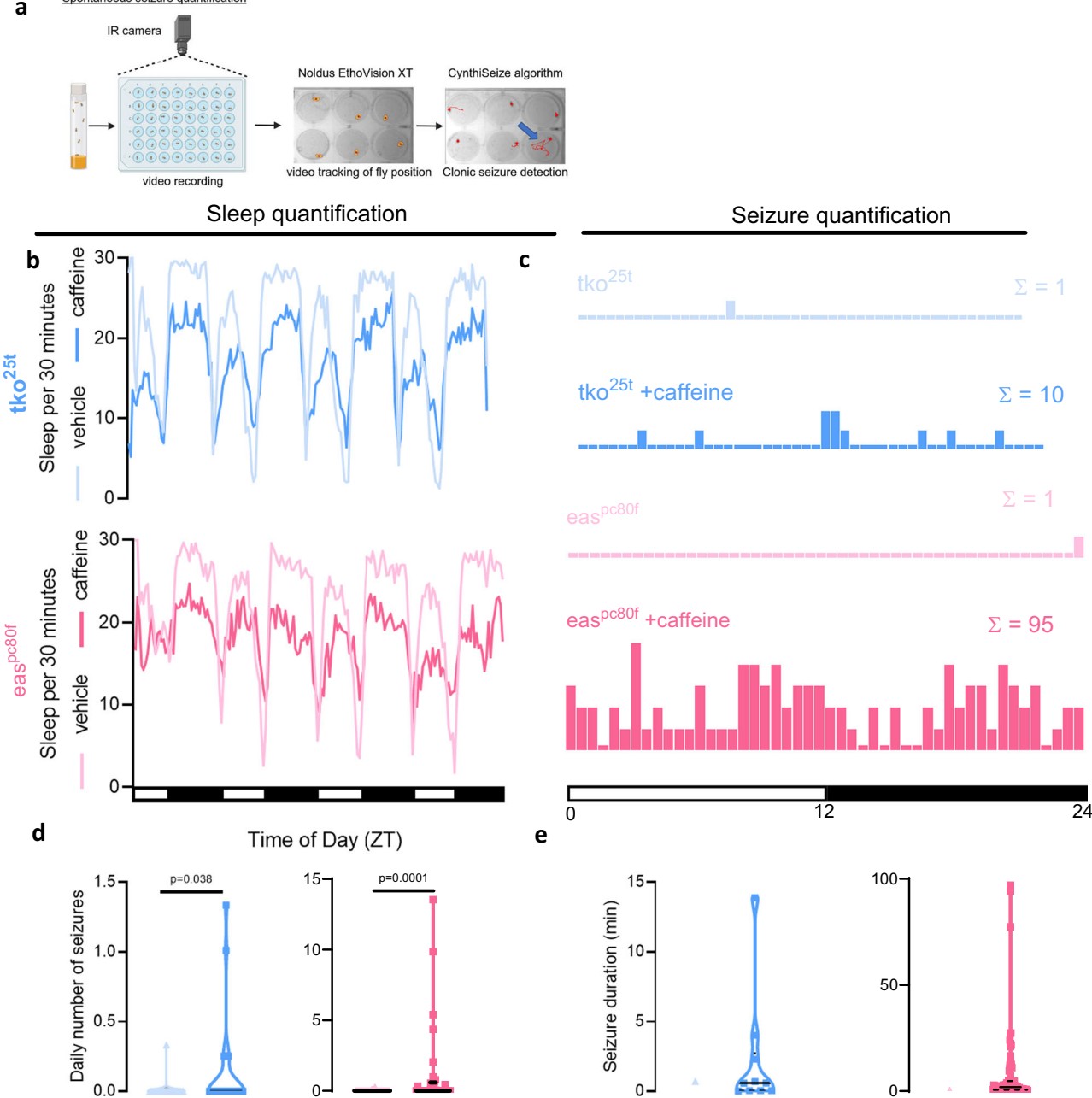

**Fig. 2 | *Drosophila* exhibit spontaneous seizures after sleep restriction. a** Flies are loaded into individual wells of a 24- or 48-well plate then video recorded for 96 h in a 12 h light:12 h dark cycle. EthoVision XT is used to detect fly position (*yellow with red dot*). An algorithm ("CynthiSeize") was developed to detect movements consistent with hyperkinetic seizures. Red line tracing = fly position over previous 10 seconds; Blue arrow = fly with sample seizure (*right*). Created in BioRender. Sehgal, A. (2025) https://BioRender.com/sxslfsb. **b** *tko*[25t] (*blue*), and *eas*[pc80f] *(pink)*, flies were fed vehicle or caffeine and sleep was measured over multiple day (white bar in *x-axis*) and night (dark bar in *x-axis*) cycles. Caffeine treatment led to less sleep across both models. n = 29–32 flies/condition. **c** Spontaneous seizures were counted across 48 30-minute time bins for each treatment condition in *tko*[25t] (*blue*) and *eas*[pc80f] (*pink*) flies. Sleep loss with caffeine

treatment increased the number of spontaneous seizures (Σ) across both genotypes. n = 29-32 flies/condition. **d** Average number of seizures per day per fly in *tko*[25t] (*blue*) and *eas*[pc80f] *(pink)* flies is increased by caffeine across genotypes. n = 31 flies/condition for *tko*[25t] flies and 32 flies/condition for *eas*[pc80f] flies. **e** Average seizure duration trends upward in *tko*[25t] (*blue*) and *eas*[pc80f] (*pink*) flies. Statistical testing not possible due to only one fly with a seizure in *tko*[25t] and *eas*[pc80f] flies without caffeine. Error bars not depicted given only one seizure detected in conditions without caffeine. With caffeine, n = 10 seizures for *tko*[25t] flies and 82 seizures for *eas*[pc80f] flies. Violin plots depict a solid line at median and dotted lines at quartiles. Negative binomial model with Wald test for daily number of seizures was used. Data are presented as mean values ± SEM. Source data are provided as a Source Data file.

Thus, the hyperkinetic episodes identified with video tracking meet the behavioral and brain-dependent definitions of seizures: (1) The hyperkinetic episodes we characterize match stereotyped tonic-clonic seizures classically identified in bang-sensitive mutant flies (compare to ref. 29). (2) These types of hyperkinetic movements do not spontaneously occur in wild-type

flies reared on normal media. (3) The hyperkinetic behavioral episodes are correlated with hypersynchronous brain activity as measured by calcium-dependent luciferase activity. (4) Seizures worsen with sleep loss as demonstrated in mammalian models and people with epilepsy. (5) Conventional anti-seizure medications inhibited seizure activity.

## Sleep loss leads to spontaneous seizures

Using chronic video-tracking, we confirmed that caffeine treatment led to sleep loss in $tko^{25t}$ and $eas^{pc80f}$ mutant flies (Fig. 2b, Supplementary Fig. 7a, e); in addition, we identified more frequent and severe spontaneous seizures. While spontaneous seizures in bang-sensitive mutants were very rare, sleep restriction with caffeine caused an increase in seizure number and duration across genotypes (Fig. 2c–e). Seizures were more likely to occur when flies were awake (Supplementary Fig. 7b, f), which correlated with decreased sleep amount in the preceding 3 hours (Supplementary Fig. 7c, g). Lethal seizures were also more likely to occur when there was decreased sleep in the preceding 3 h (Supplementary Fig. 7d, h). These data confirm that bang-sensitive, neurogenetic models of fly epilepsy exhibit spontaneous seizures, and that spontaneous lethal seizures are more likely to occur following prolonged wakefulness.

Beyond neurogenetic forms of epilepsy, we sought to understand if the effects of sleep restriction on seizures were generalizable to other forms of seizures. We fed picrotoxin, a GABA$_A$ receptor antagonist to wild-type Canton-S flies and performed continuous video tracking (see Supplementary Movie 2, 3, 4). Even in the absence of a sleep-depriving stimulus, retrospective analyses revealed that seizures were more likely to be lethal when individual flies were awake and had slept less in the preceding three hours (Supplementary Fig. 8a–c). Seizures were more frequent during wakefulness (Supplementary Fig. 8d), possibly due to decreased recent sleep (Supplementary Fig. 8e). Combined with the results from neurogenetic models of seizures, these data indicate that the total amount of sleep achieved and/or the drive for sleep may be modulators of seizure severity.

## Sleep drive, not total sleep time, controls seizure severity

To disambiguate the effects of sleep history (i.e. how much an organism has slept) versus sleep drive (i.e. how much sleep an organism requires) on seizure severity, we took advantage of four thermogenetic models of sleep loss that have variable effects on sleep drive. In the $tko^{25t}$ background, we drove expression of UAS-TrpA1, a temperature-sensitive cation channel, using the c584-Gal4 and Tdc2-Gal4 drivers. c584 labels a diverse group of cells, including dopaminergic neurons, while Tdc2 targets octopamine-producing neurons. With temperature elevation, c584>TrpA1 leads to sleep loss followed by a homeostatic increase in sleep, indicating that sleep drive has accumulated[30]. We herein refer to the c584>TrpA1 group as a "sleep rebound group". Specifically, in the "sleep rebound group", increasing temperature from 18 °C to 30 °C for 12–24 h decreased sleep (Fig. 3b, c) and subsequently worsened seizure severity (Fig. 3f) predominantly driven by a prolongation of the convulsive phase (as detailed in *Supplementary Methods* flies were brought to room temperature for 20 min prior to seizure testing) (Supplementary Fig. 9g–i). After returning the temperature to 18 °C, c584>TrpA1 flies exhibited increased nighttime rebound sleep (Fig. 3d) and prolonged sleep bouts (Supplementary Fig. 9a–d), consistent with a homeostatic sleep recovery. In contrast, Tdc2>TrpA1 is known to lead to sleep loss without homeostatic rebound sleep[31]. With the manipulation that increases wake but not rebound, the same temperature shift also reduced sleep (Fig. 3g, h), but here there were minimal effects on seizures (Fig. 3k and Supplementary Fig. 9n–r). After the temperature was shifted back from 30 °C to 18 °C, this "no sleep rebound group" continued to demonstrate decreased sleep and no indication of homeostatic sleep rebound or consolidation (Fig. 3i and Supplementary Fig. 9j–m) as previously described[31]. Calculation of p(Doze), a metric of sleep pressure/sleep drive[32], demonstrated that the "sleep rebound group" exhibits increased sleep drive (Fig. 3e), but the "no sleep rebound group" does not (Fig. 3j).

To further extend these findings, we activated two additional subsets of neurons labeled by the c453-Gal4 and 104906-Gal4 drivers that have previously been demonstrated to lead to sleep loss[30]. TrpA1-mediated activation of neurons expressing c453-Gal4, which leads to sleep loss without sleep rebound[30], does not worsen seizures (Supplementary Fig. 10). In contrast, activation of TrpA1 driven by 104906-Gal4, which leads to sleep loss followed by sleep rebound[30], exacerbates seizures (Supplementary Fig. 10). To test a more naturalistic stimulus of sleep loss that does not lead to sleep rebound[33], we starved flies for 12 h and found that seizures are not significantly worsened (Supplementary Fig. 11). In conjunction with the effects of sleep drive on spontaneous seizures, these experiments implementing thermogenetic- and starvation-induced sleep loss consistently indicate that manipulations increasing sleepiness, but not sleep amount per se, drive seizure severity.

## Sleep-promoting circuits modulate seizures

The above data indicate that sleep drive might have direct effects on seizure severity independent of sleep amount. To test this, in the $tko^{25t}$ background we expressed UAS-TrpA1 in 12 cellular populations or circuits that are known to play a role in wakefulness or sleep. Broad Gal4 drivers were selected for the purposes of screening. Flies were raised at 18 °C to minimize TrpA1 activation then brought to 25 °C for TrpA1 activation (Fig. 4a). This activation temperature was selected because it adequately increases TrpA1 conductance[34] and also avoids direct effects of temperature on bang-sensitive seizures in the $tko^{25t}$ background[29]. After bang-sensitive seizure induction, flies were then returned to 25 °C for video recording and seizure quantification (Fig. 4a). Importantly, in these experiments, discrete cell populations and circuits involved in the sleep/wake cycle were activated for 4 min only, without allowing sufficient time for sleep amount to increase or decrease. Given that sleep here is defined as immobility for at least 5 min[35], total sleep amount was unchanged. This allowed us to test direct effect of the cells/circuits of interest on seizure severity independent of sleep behavior.

Using this paradigm, we found that acute direct activation of sleep-promoting circuits worsened seizure severity. As compared to genetic controls, activation of 104 y>TrpA1 (expressed in the dorsal fan-shaped body (dFB)), 201 y>TrpA1 (expressed in the mushroom body (MB) as well as 50–100 non-MB cells per hemisphere)[36], and kurs58>TrpA1 (expressed in the pars intercerebralis (PI)) prolonged the tonic/clonic phase of seizures (Fig. 4b, c). The dorsal fan-shaped body, mushroom body, and pars intercerebralis each have known sleep-promoting functions, as previously reviewed[37]. In addition to drivers expressed in the dFB, MB, and PI, R58H05>TrpA1, expressed in the ellipsoid body (EB), another sleep-promoting region of the brain, increased total seizure times (Fig. 4d, e). While c584>TrpA1 activation reduced tonic/clonic times (note that this protocol differed from the one above in not allowing sleep decrease), other wake-promoting drivers including 11H05>TrpA1 and 60D04>TrpA1 were not protective (Fig. 4b, c). The only neurotransmitter/neuromodulator group that worsened seizure severity upon acute activation were *trh*-expressing serotonergic neurons (Fig. 4d, e). None of the activated populations prolonged recovery times (Supplementary Fig. 12c), suggesting that other brain regions may be involved in regulation of postictal seizure recovery. In summary, these data demonstrate that acute activation of brain regions that encode sleep drive, including the dFB and MB, worsens seizures, even with no change in total sleep amount.

## Sleep loss increases activity of sleep-promoting circuits

These thermogenetic studies implicating the activity of sleep-promoting circuits in seizure severity suggested that sleep-promoting circuits change activity of other brain regions. To address this question, we investigated the effect of sleep loss on brain wide activity by pan-neuronally expressing UAS-CaLexA, a genetically-encoded, fluorescent calcium indicator as a surrogate readout for brain activity. After maintenance of flies on caffeine for 48 h to restrict sleep, we dissected whole brains and measured endogenous CaLexA

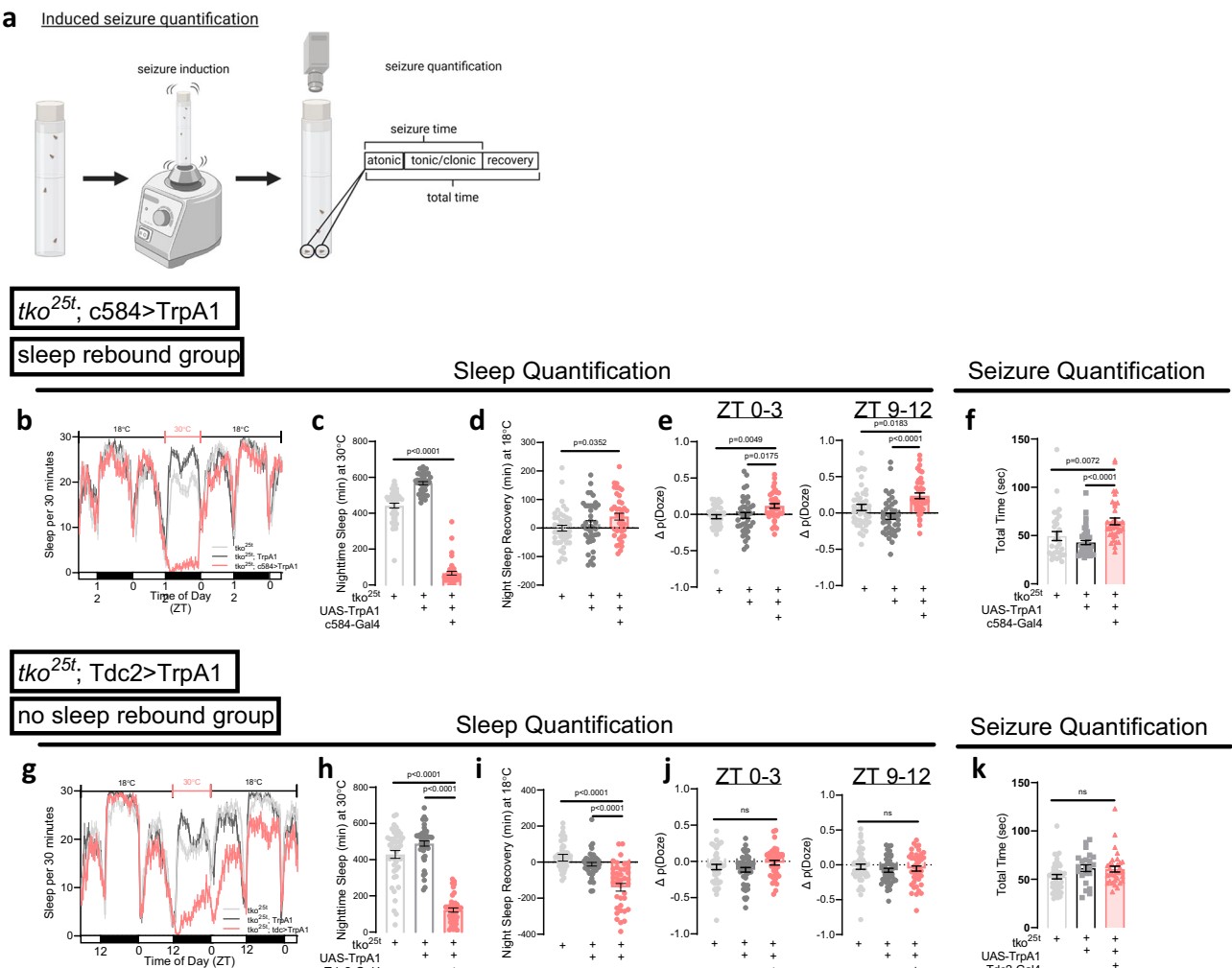

**Fig. 3 | Sleep loss associated with increased sleep drive exacerbates seizures.**
**a** Experimental protocol depicting seizure induction on vortexer followed by video recording of fly seizures. Seizures are quantified for previously detailed seizure phases. Created in BioRender. Sehgal, A. (2025) https://BioRender.com/p4jrytx. **b**–**d** *tko²⁵ᵗ*; c584-Gal4>UAS-TrpA1 flies, the "sleep rebound group", were maintained at 18 °C, shifted to 30 °C overnight for 12 h, then recovered in 18 °C. Thermogenetic activation of the "sleep rebound group" at 30 °C led to sleep loss that recovered back to baseline at 18 °C. After returning to 18 °C, night sleep was increased as compared to baseline night sleep, consistent with homeostatic sleep rebound. *ZT* = zeitgeber time. **e** Change in p(Doze) was calculated at ZT 0-3 and ZT 9-12 and reveals higher levels in the "sleep rebound group" consistent with increased sleep drive. For (**b**–**e**), n = 43 for *tko²⁵ᵗ* flies and n = 40 flies for *tko²⁵ᵗ*;UAS-TrpA1 and *tko²⁵ᵗ*; c584-Gal4>UAS-TrpA1 each. **f** After thermogenetic activation of the "sleep rebound group" for 24 h, total seizure times are prolonged. n = 30 *tko²⁵ᵗ* flies, n = 45 *tko²⁵ᵗ*;

UAS-TrpA1 flies, and n = 38 *tko²⁵ᵗ*; c584-Gal4>UAS-TrpA1 flies. **g**–**i** *tko²⁵ᵗ*; Tdc2-Gal4>UAS-TrpA1 flies, the "no sleep rebound group", were maintained at 18 °C, shifted to 30 °C overnight for 12 h, then recovered in 18 °C. Thermogenetic activation of the "no sleep rebound group" at 30 °C led to sleep loss that persisted even after flies were restored to 18 °C. There was no homeostatic sleep rebound after temperature was returned to 18 °C. **j** Change in p(Doze) was calculated for the "no sleep rebound group" and demonstrates no significant changes, indicating no change in sleep drive with sleep loss. For *g-j*, n = 45 for *tko²⁵ᵗ* flies, n = 46 for *tko²⁵ᵗ*;UAS-TrpA1 flies, and n = 41 for *tko²⁵ᵗ*;Tdc2-Gal4>UAS-TrpA1 flies. **k** After thermogenetic activation of the "no sleep rebound group" for 24 h, there was no significant difference in total seizure times. n = 45 for *tko²⁵ᵗ* flies, n = 22 for *tko²⁵ᵗ*;UAS-TrpA1 flies, and n = 32 for *tko²⁵ᵗ*;Tdc2-Gal4>UAS-TrpA1 flies. One-way ANOVA with Tukey's multiple comparisons test was used. Data are presented as mean values ± SEM. Source data are provided as a Source Data file.

signal. Consistent with previous findings, we found increased intracellular calcium in sleep-promoting brain regions including the dFB and EB[7,9], as well as the MB (Fig. 5a, c–e). Incredibly, beyond regions of the brain known to be sleep-promoting, we found that sleep loss induced by caffeine led to brain-wide increases in intracellular calcium (Fig. 5b), indicating that brain-wide activity may be increased in the setting of sleep loss. These effects were not specific to caffeine, as (1) mechanical sleep deprivation and (2) high sleep pressure at the end of the day (ZT12) also led to increased calcium in sleep-promoting brain regions (Supplementary Fig. 13a–c). These data demonstrate that sleep loss leads to increased activity of sleep-promoting circuits, and this correlates with increased brain-wide activity. Broad neural hyperactivity predisposes the brain to seizures,

and increases in electrical activity have been previously observed after hyperactivation of sleep-promoting neurons[38]. Therefore, we sought to understand if activity of sleep-promoting circuits broadly modulates seizure risk.

**Circuits encoding sleepiness can toggle seizures**
We next asked if direct activation of sleep-promoting circuits increases the likelihood and severity of spontaneous seizures, as it does for bang-sensitive seizures. To achieve broad control of sleep-promoting circuits, we turned to our chronic video tracking paradigm and expressed UAS-csChrimson under control of two largely non-overlapping sleep-promoting drivers–23E10-Gal4, which is expressed in the dFB and VNC[7,8,15,16], and 201y-Gal4, expressed in the MB and

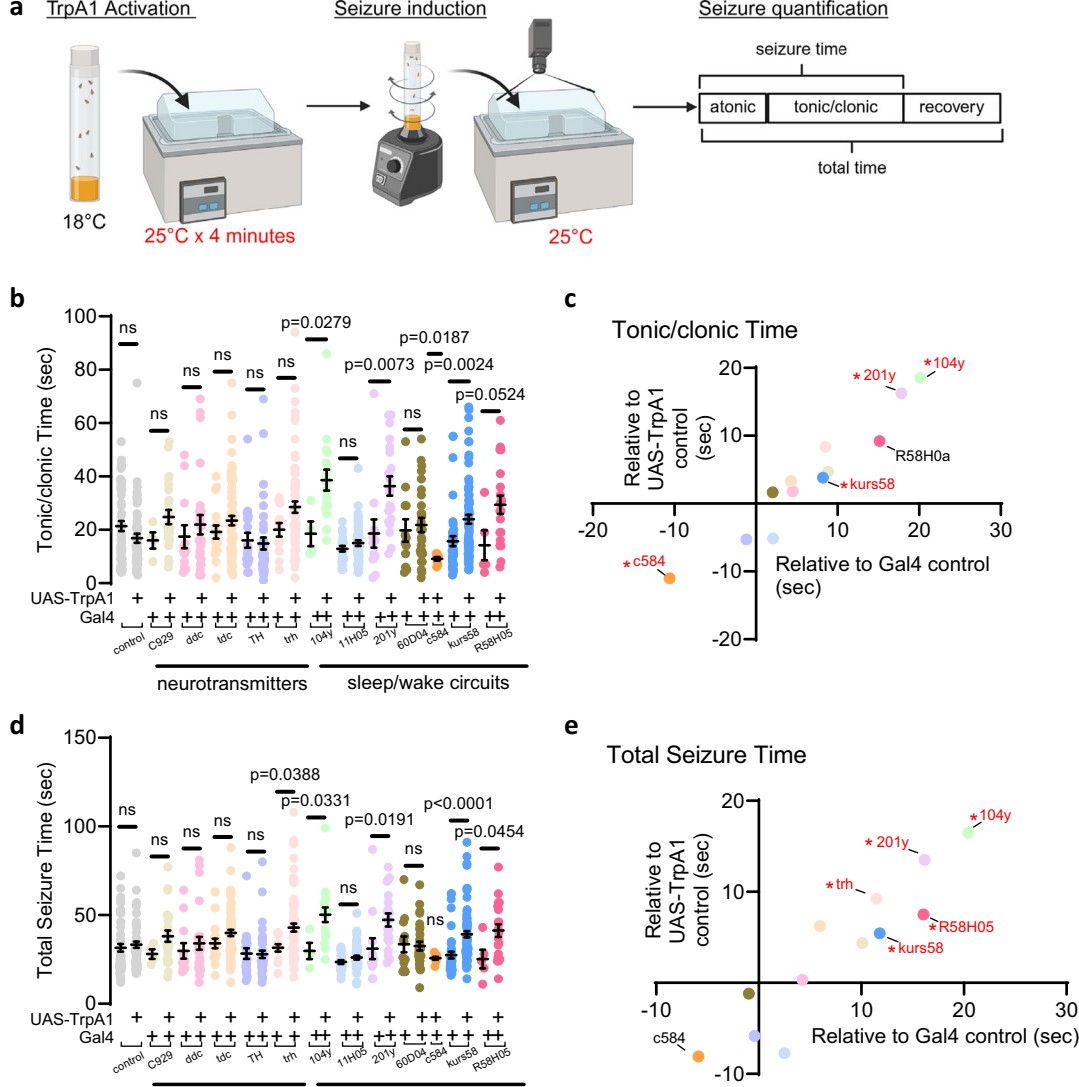

**Fig. 4 | A Gal4 screen reveals activation sleep-promoting circuits worsens seizures. a** Experimental protocol depicting flies in the $tko^{25t}$ background were raised at 18 °C, placed at 25 °C for 4 min for TrpA1 activation, vortexed for seizure induction, and video recorded for seizures at 25 °C. Atonic, tonic/clonic, and recovery phases of seizures were quantified. To drive TrpA1, Gal4 drivers were selected for peptidergic (C929-Gal4), serotonergic/dopaminergic (Ddc-Gal4), octopaminergic (Tdc2-Gal4), dopaminergic (TH-Gal4), serotonergic (trh-Gal4), dorsal fan-shaped body (104y-Gal4), wake-promoting (11H05-Gal4, 60D04-Gal4, c584-Gal4), mushroom body (201y-Gal4), pars intercerebralis (kurs58-Gal4), and ellipsoid body (R58H05-Gal4) neurons. Created in BioRender. Sehgal, A. (2025) https://BioRender.com/pt2kiq2. **b, c** Tonic/clonic times were prolonged upon driving of TrpA1 with 104y-Gal4, 201y-Gal4, kurs58-Gal4, and improved with c584-Gal4. **d, e** Total seizure times

demonstrate worsening of seizures upon driving of TrpA1 with 104y-Gal4, 201y-Gal4, kurs58-Gal4, R58H05-Gal4, and trh-Gal4. **c, e** The *x* values represent experimental seizure durations minus the seizure durations in the UAS-TrpA1 controls. The *y* values represent experimental seizure durations minus seizure durations in the Gal4 genetic controls. Significant values (p<0.05) values are *red* with an asterisk. The number flies assessed for each condition is given in (): $tko^{25t}$ (52), TrpA1 (52), C929 (4), C929>TrpA1 (23), ddc (12), ddc>TrpA1 (26), tdc (17), tdc>TrpA1 (69), TH (19), TH>TrpA1 (39), trh (12), trh>TrpA1 (68), 104 y (4), 104 y>TrpA1 (17), 11H05 (22), 11H05>TrpA1 (51), 201 y (12), 201 y>TrpA1 (20), 60D04 (12), 60D04>TrpA1 (28), c584>TrpA1 (8), kurs58 (41), kurs58>TrpA1 (77), R58H05 (5), and R58H05>TrpA1 (20). Two-group two-tailed t-test. Data are presented as mean values ± SEM. Source data are provided as a Source Data file.

50–100 non-MB cells per hemisphere. Flies were raised in the dark, entrained in blue light and fed ATR for 48 h, then placed into individual wells of a 24–48-well plate containing picrotoxin with stimulation of UAS-csChrimson using red light for the first 5 min of every hour (Supplementary Fig. 14a). Experimental flies and genetic controls were also fed caffeine to test if sleep restriction increases seizures by increasing activity of these sleep-promoting regions, in which case it would have little additional effect, or if it acts synergistically with activation of sleep-promoting regions. As expected, caffeine treatment of genetic controls (23E10-Gal4, 201y-Gal4 + caf) led to sleep loss (Supplementary Fig. 14b, d) and more frequent and severe seizures (Supplementary Fig. 14c, f, g).

Prolonged activation of sleep-promoting cells with csChrimson (23E10-Gal4, 201y-Gal4>csChrimson) led to increased sleep (Supplementary Fig. 14e). This did not worsen seizure severity (Supplementary Fig. 14c, f, g), likely because although sleep drive was induced, flies had an opportunity to dissipate their sleep drive by sleeping more (Supplementary Fig. 14b, e), thereby reducing their seizure risk. In an effort to induce sleepiness without allowing flies sufficient time to dissipate their sleepiness by sleeping more, we attempted two additional csChrimson stimulation protocols: (1) red light 2 min on and 3 min off, and (2) red light on continuously. The former protocol enhanced sleep of 23E10-Gal4, 201y-Gal4>csChrimson flies even more, while the constant red-light exposure in the latter protocol disrupted seizure

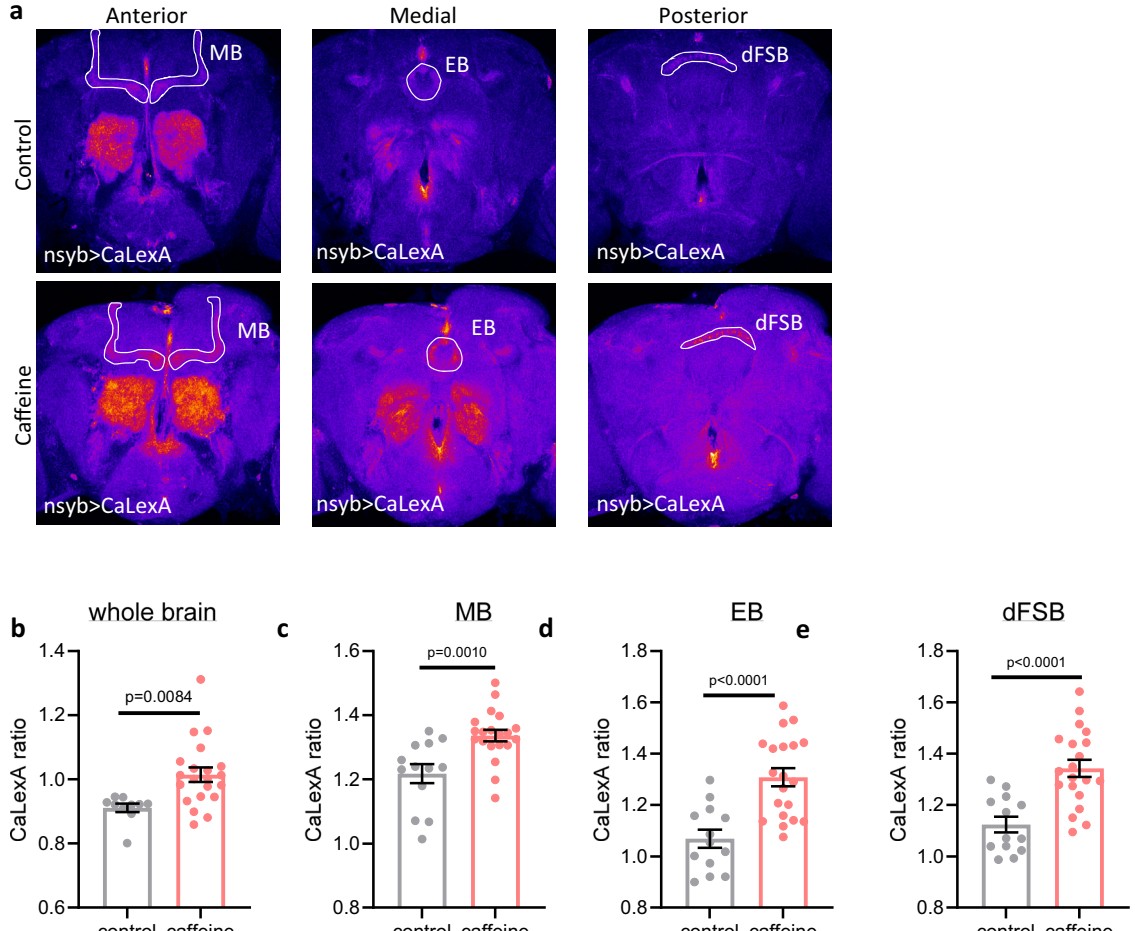

**Fig. 5 | Activity is increased across whole brain, including sleep-promoting circuits, after sleep restriction.** Brains dissected from nsyb-Gal4>CaLexA flies treated with vehicle or caffeine. **a** Representative images in the anterior, medial, and posterior brain revealing the mushroom body (MB), ellipsoid body (EB), and dorsal fan-shaped body (dFSB). White lines outline the region of interest used for subregion quantification. Images are pseudocolored with the "fire" lookup table with warmer colors indicating increased intensity. **b**–**e** Sleep restriction with caffeine in nsyb-Gal4>CaLexA flies leads to increased whole brain, MB, EB, and dFSB CaLexA (GFP:RFP) signal. n = 10-21 flies/condition. Two-group two-tailed t-test. Data are presented as mean values ± SEM. Source data are provided as a Source Data file.

initiation even in control conditions for unclear reasons. Therefore we returned to our initial stimulation protocol using stimulation for the first 5 min of every hour (Supplementary Fig. 14a). When sleep-promoting cells were activated with csChrimson (23E10-Gal4, 201y-Gal4>csChrimson flies) with simultaneous caffeine administration, there was no opportunity to dissipate sleepiness (Supplementary Fig. 14b, d, e); in these flies, seizure frequency and severity were similar to those of genetic controls (23E10-Gal4, 201y-Gal4) fed caffeine (Supplementary Fig. 14c, f, g). Importantly, caffeine treatment with simultaneous activation of sleep-promoting brain regions (23E10-Gal4, 201y-Gal4>csChrimson flies) led to seizures, including lethal seizures, even though there was increased sleep per fly in the 3 h preceding seizures as compared to genetic controls (Supplementary Fig. 15c, d). This suggests that during aberrant hyperactivation of sleep-promoting brain regions, seizures continue to occur even if individual flies increase total sleep amount.

If hyperactivation of sleep-promoting circuits increases seizure severity (Supplementary Fig. 14c, f), then this raised the exciting possibility that inactivation of sleep-promoting circuits might be a strategy to decrease seizure risk. For inhibition, we expressed UAS-GtACR1, a light-sensitive anion channelrhodopsin in sleep-promoting cells using 23E10-Gal4 and 201y-Gal4. Similar to the protocol for csChrimson activation, we raised flies in the dark then entrained flies for 48 h in red light with ATR, then placed them into 24-48-well plates containing picrotoxin (Fig. 6a). While caffeine treatment of genetic controls

(23E10-Gal4, 201y-Gal4 + caf) led to decreased sleep (Fig. 6b, d, e) and increased seizures (Fig. 6c, f, g), experimental inactivation of sleep-promoting brain regions (23E10-Gal4, 201y-Gal4 > GtACR1 flies) with caffeine administration reduced sleep (Fig. 6b, d, e) and also decreased seizures (Fig. 6c, f, g). Inhibition of sleep-promoting cells (23E10-Gal4, 201y-Gal4 > GtACR1 flies) in vehicle conditions also led to decreased sleep, comparable to caffeine treatment, with fewer and less severe seizures (Fig. 6b, d, e, Supplementary Fig. 15f–h). Reminiscent of the effects of the "no sleep rebound group" (Tdc2>TrpA1 and c453>TrpA1 flies) and starvation noted above, this reduced sleep amount is presumably associated with decreased sleep drive secondary to direct inactivation of sleep-promoting cells with GtACR1. These optogenetic experiments provide direct evidence that the effects of sleep drive on seizure risk can be dissociated from the effects of sleep amount.

## Loss of 5HT1A mediates increased seizures

To understand how sleep-promoting circuits might become hyperactive to increase seizure burden, we performed RNA sequencing of the dFB with and without sleep deprivation. Given the ability of the dFB to affect whole-brain activity, we focused our attention on transcripts encoding neurotransmitter or neuromodulator receptors and rate limiting enzymes (Fig. 7a, Supplementary Fig. 16). Of these transcripts, only one significantly changed after sleep restriction: 5HT1A. The serotonin 5HT1A receptor is a G protein-coupled receptor that couples with inhibitory $G_i$ proteins as previously reviewed[39]. Downregulation of

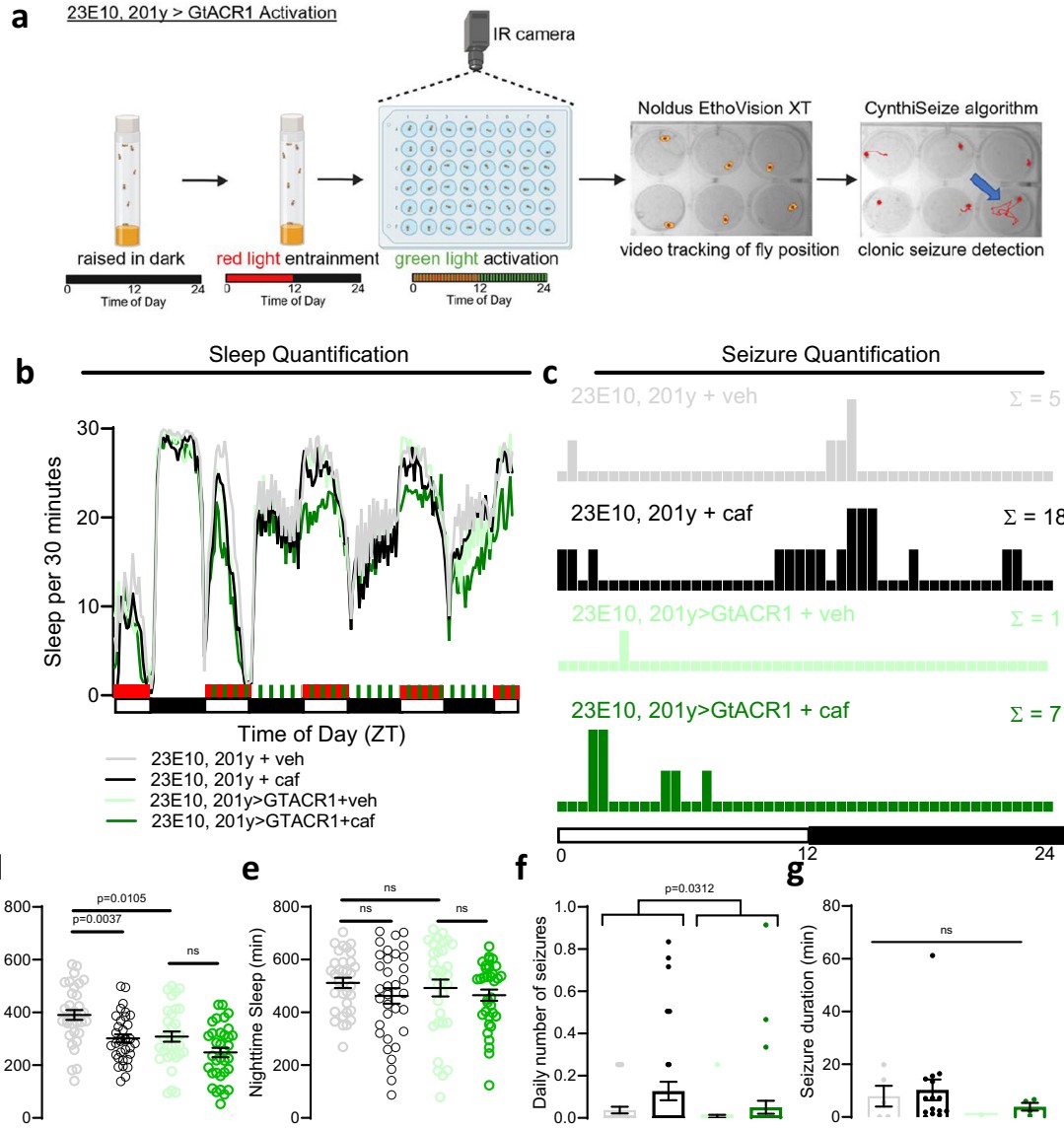

**Fig. 6 | Inhibition of sleep-promoting circuits protects against seizures in the setting of sleep loss. a** Experimental protocol showing flies were raised in darkness, entrained in red light, then placed into 24- or 48-well plates for chronic video monitoring with green light stimulation. Fly positions over time were converted into XY coordinates, and a "CynthiSeize" algorithm was developed to identify seizures. Created in BioRender. Sehgal, A. (2025) https://BioRender.com/9siy0zu. **b, d, e** Sleep quantification after 23E10-Gal4, 201y-Gal4 > GtACR1 activation. Caffeine and 23E10-Gal4, 201y-Gal4 > GtACR1 decrease sleep. n = 34 flies/condition.

**c, f, g** Seizure quantification after 23E10-Gal4, 201y-Gal4 > GtACR1 activation. Caffeine increases seizure frequency and this is blocked with 23E10-Gal4, 201y-Gal4 > GtACR1 activation. n = 34 flies/condition. One-way ANOVA with Dunnett's T3 multiple comparisons adjustment, Kruskal-Wallis test with Dunn's multiple comparisons adjustment, negative binomial model with Wald test (for daily number of seizures), or mixed effects model (for seizure durations) was used. Data are presented as mean values ± SEM. Source data are provided as a Source Data file.

5HT1A in the dFB after sleep restriction (Fig. 7a) would be predicted to decrease inhibitory signaling, thereby leading to increased activity of the dFB and enhanced sleep pressure.

We hypothesized that activation of 5HT1A in sleep-promoting circuits would decrease sleep. To test this hypothesis, we used our chronic video tracking platform (Fig. 2a) and induced seizures with picrotoxin. As hypothesized, genetic control flies (23E10-Gal4, 201y-Gal4) fed 8-OH-DPAT, a selective 5HT1A receptor agonist, exhibited decreased sleep (Fig. 7b, d). Although there was decreased sleep, there was a decrease in seizure frequency (Fig. 7c, f). This likely occurs because 5HT1A inhibits sleep-promoting cells, thereby decreasing both sleep amount and sleep drive. To better understand the site of action of 8-OH-DPAT, we knocked down expression of 5HT1A in sleep-promoting cells using RNAi (23E10-Gal4, 201y-Gal4 >

5HT1A RNAi). Knockdown of 5HT1A in sleep-promoting cells caused an increase in sleep (Fig. 7b, d). This is consistent with the RNAseq results (Fig. 7a, Supplementary Fig. 16), as sleep restriction leads to loss of 5HT1A expression, thereby driving increased activity of the dFB and leading to more sleep. The increased sleep resulting from loss of 5HT1A expression in the dFB and MB would lead to dissipation of sleep drive. Consistent with this possibility, we found loss of 5HT1A expression led to decreased seizure risk (Fig. 7c, f, g). When 8-OH-DPAT was fed to flies with 5HT1A RNAi expressed in sleep-promoting brain regions (23E10-Gal4, 201y-Gal4 > 5HT1A RNAi flies), it was significantly less effective at reducing sleep with an effect size consistent with the reduced 5HT1A function caused by this RNAi line[40,41]. This indicates the effects of 5HT1A agonism is mostly occurring through its expression in sleep-promoting cells (Fig. 7d, e).

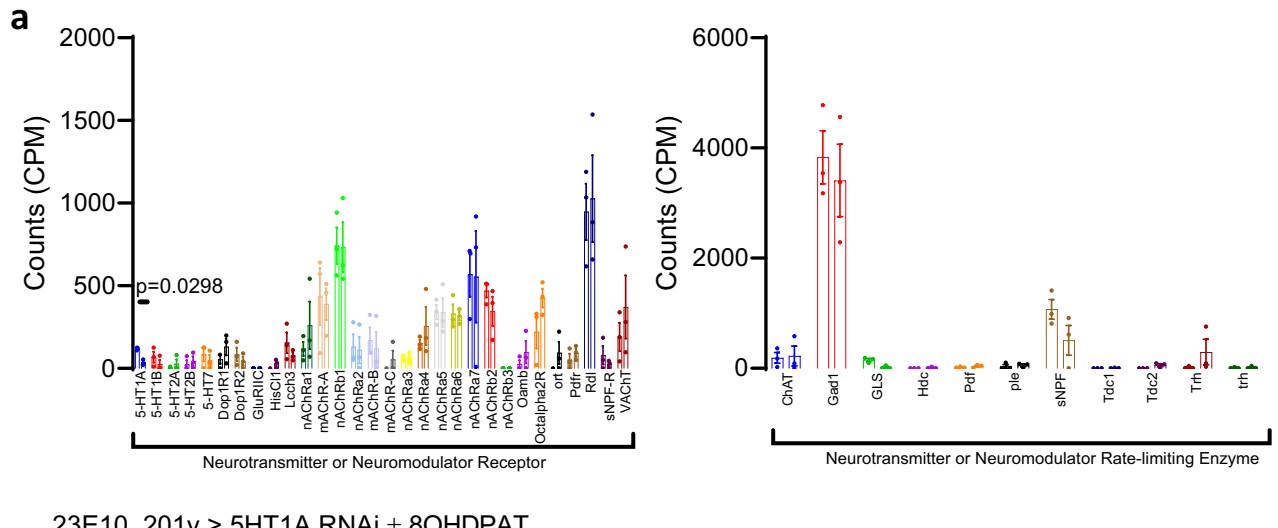

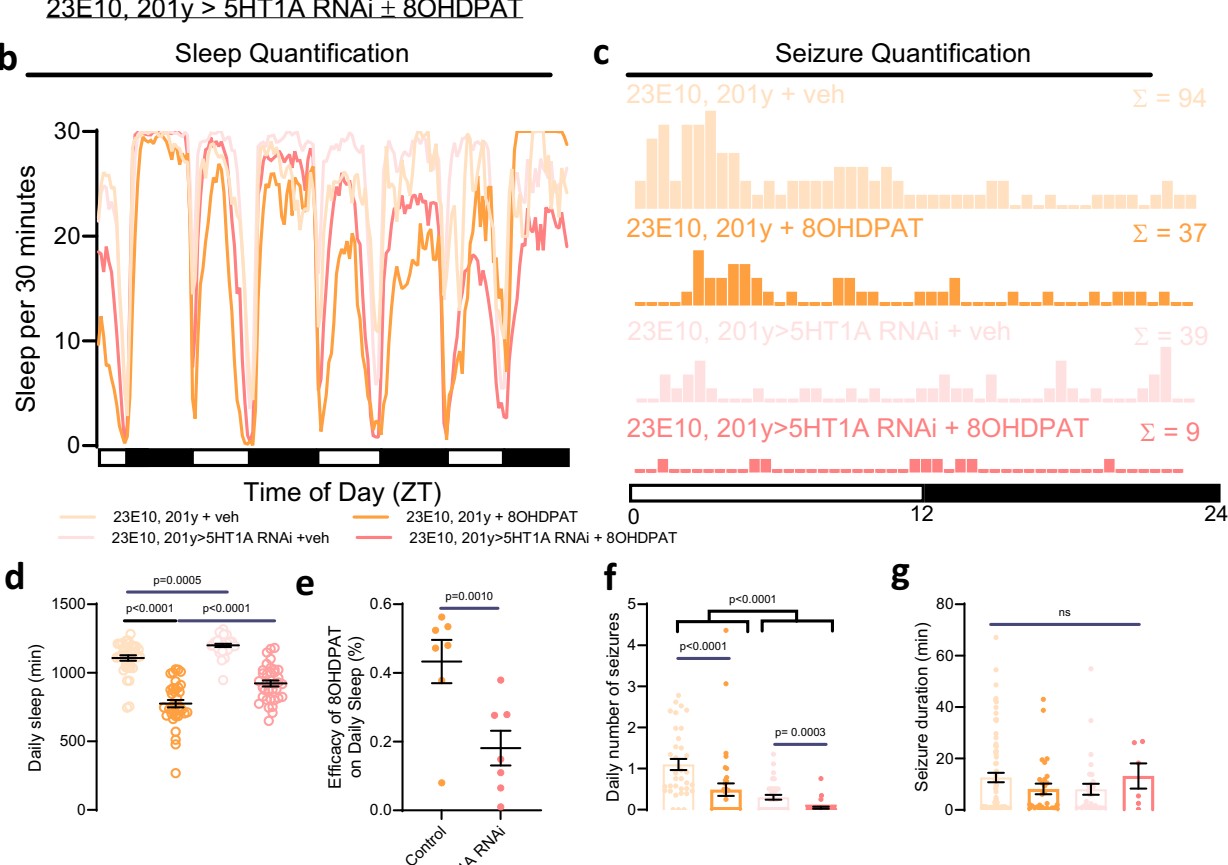

**Fig. 7 | 5HT1A is downregulated after sleep loss, and this is sleep-promoting.**
**a** Transcriptomic analysis of the dorsal fan-shaped body after sleep-restriction reveals downregulation of 5HT1A, but not other neurotransmitter receptors or rate-limiting enzymes. N = 3 biological replicates. **b**, **d** Sleep quantification after addition of 8-OH-DPAT, a selective 5HT1A receptor agonist, reveals decreased sleep, consistent with the inhibitory effects of 5HT1A on the activity of sleep-promoting centers. Conversely, 23E10-Gal4, 201y-Gal4 > 5HT1A RNAi-mediated downregulation of 5HT1A in sleep-promoting centers leads to increased sleep. n = 46 flies/condition. **e** The effect of a 5HT1A agonist on daily sleep loss is decreased after downregulation

of 5HT1A in sleep-promoting centers. n = 7 biological replicates. **c**, **f**, **g** Seizure severity is reduced after both 5HT1A agonism and 5HT1A downregulation. For daily seizure quantification, n = 36 flies/condition. For seizure duration, n = 76 seizures for '23E10, 201 y + veh', n = 29 seizures for '23E10, 201 y + 8OHDPAT', n = 30 seizures for '23E10, 201 y > 5HT1A RNAi + veh', and n = 6 seizures for '23E10, 201 y > 5HT1A RNAi + 8OHDPAT'. One-way ANOVA with Dunnett's T3 multiple comparisons test, paired two-tailed t-test, negative binomial model with Wald test (for daily number of seizures), or mixed effects model (for seizure durations) was used. Data are presented as mean values ± SEM. Source data are provided as a Source Data file.

In summary, these results demonstrate that the loss of 5HT1A-mediated inhibitory signaling promotes sleep. We also find that agonism of 5HT1A can be used to decrease sleep pressure and protect against seizures.

Given that increased 5HT1A signaling reduced sleep pressure and protected against seizures, we asked if a clinically available, Food and Drug Administration (FDA)-approved medication might be able to reverse the effects of sleepiness on seizures. Buspirone has high

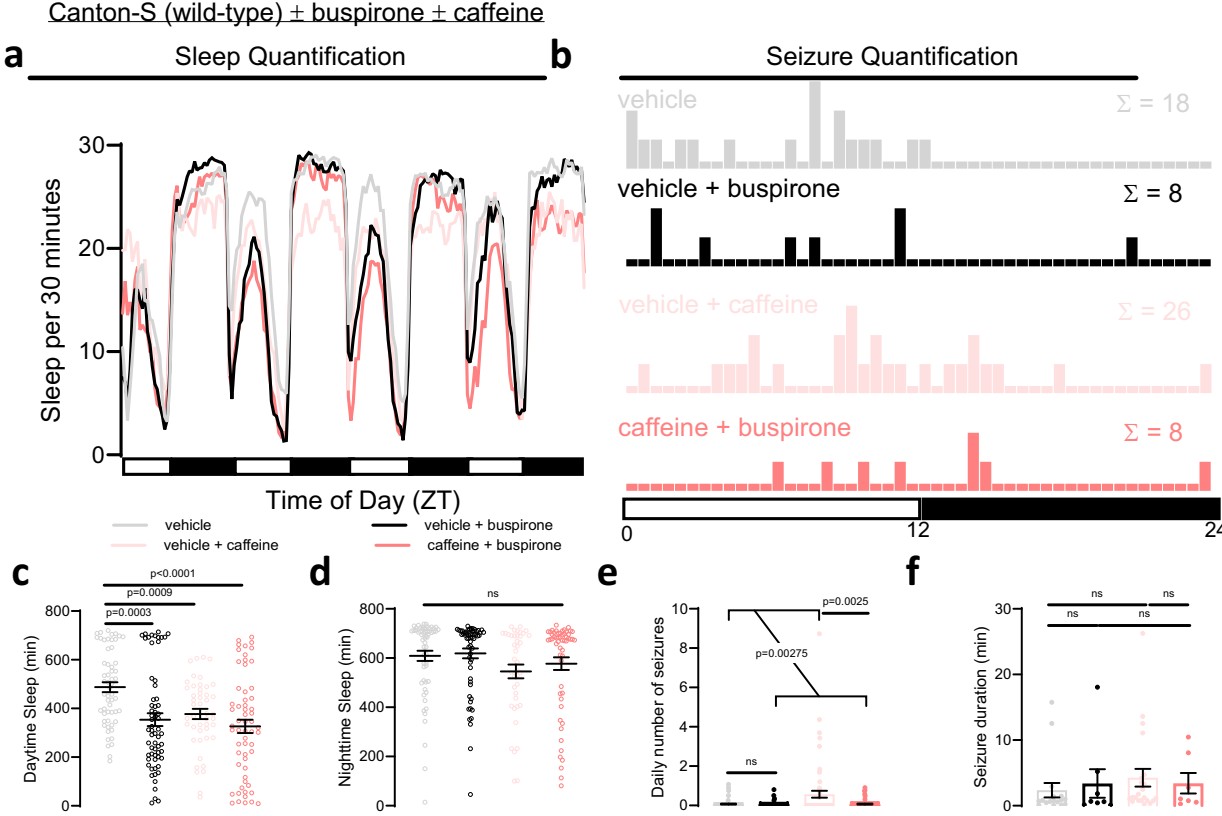

**Fig. 8 | Increasing 5HT1A activity with the FDA-approved drug buspirone can protect against seizures induced by sleep loss. a, c, d** Sleep quantification after administration of caffeine demonstrates decreased sleep, but no significant change with administration of buspirone, an FDA-approved 5HT1A agonist. n = 36 flies/condition. **b, e, f** Seizure severity is increased with caffeine, but reduced to baseline levels with co- administration of buspirone. n = 36 flies/condition. One-way ANOVA with Dunnett's T3 multiple comparisons test, paired two-tailed t-test, negative binomial model with Wald test (for daily number of seizures), or mixed effects model (for seizure durations) was used. Data are presented as mean values ± SEM.

affinity for the serotonin 5HT1A receptor, whereby it acts as a partial agonist, and is clinically approved to treat mood disorders. Consistent with the inhibitory effects of 5HT1A on sleep-promoting circuits described above, a side-effect of buspirone is insomnia[42]. We fed wild-type flies buspirone and caffeine to determine if buspirone could decrease sleep drive through 5HT1A agonism, thereby reducing the increased seizure burden associated with sleep loss. Buspirone reduced baseline sleep amount as did 8-OH-DPAT (Fig. 8a, c, d). As previously seen, caffeine led to decreased sleep and increased seizures (Fig. 8a–f). The addition of buspirone to caffeine did not significantly affect caffeine-induced sleep loss (Fig. 8a, c, d), but it did decrease seizure risk (Fig. 8b, e). Thus, we find that buspirone, an FDA-approved 5HT1A agonist, reverses the increased seizure burden occurring after sleep restriction.

## Discussion

Our data suggest that after sleep loss, the primary cause of worsened seizures is sleep drive, and not sleep amount. We use eight complementary methods to restrict sleep [(1) caffeine, (2) two short-sleeping mutants, thermogenetic wake-promoting neuron activation with (3) c584>TrpA1, (4) Tdc2>TrpA1, (5) c453>TrpA1, and (6) 104906>TrpA1, (7) starvation, and (8) optogenetic sleep-promoting neuron inhibition with 23E10, 201 y > GtACR1] and consistently find when sleep drive is increased, seizures are worsened. Using multiple modalities of sleep restriction provided confirmation that the observed effects of sleep loss on seizures were not simply an artifact of any one manipulation. Manipulations that sustain activity of sleep-promoting centers increase

seizure frequency. Importantly, the converse is also true: if excitation of sleep-promoting centers is decreased, then seizures are less severe, even in the setting of sleep loss. We propose that targeting neural correlates of sleepiness can improve seizure control (Fig. 9).

In considering the effects of sleep on seizures, there are multiple variables to account for, including, but not limited to, (1) sleep history, which is the total sleep amount prior to an event of interest, (2) sleepiness, which is the drive to sleep more, (3) sleep or wake status at the time of an event of interest, and (4) how recently a state change (wake → sleep or sleep → wake) occurred. Our video tracking platform allowed for quantification of these variables while simultaneously monitoring for seizures in *Drosophila*. We present multiple lines of evidence demonstrating that the drive to sleep has a greater impact on seizure burden than sleep amount. These results provide an explanation to a discrepancy in clinical neurology: while individuals with epilepsy identify decreased sleep as a trigger for seizures[2,3], objective measurement of sleep amount in individuals with epilepsy demonstrates a poor correlation with seizure risk[43]. Given sleep drive is variable between individuals[44], our data suggest that increasing sleep quantity is not key to preventing seizures, but rather providing sufficient rest to minimize sleep drive decreases seizure risk.

Given that seizures are not generated in sleep-promoting circuits, we sought to understand how sleep drive affects activity across the brain. Using intracellular calcium concentration as a proxy for neuronal activity, we confirmed that elevated sleep pressure is associated with increased activity of sleep-promoting regions of the brain (Fig. 5, Supplementary Fig. 13), consistent with previous findings[7–9]. We find

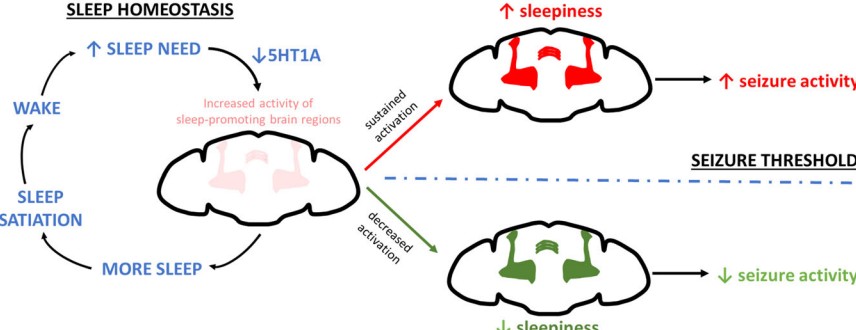

**Fig. 9 | Schematic depicting how homeostatic sleep drive drives increased seizure risk.** Sleep drive leads to increased activity of sleep-promoting circuits, and we find that this, in part, is driven by decreased expression of 5HT1A in the dorsal fan-shaped body (dFB). Increased activity of sleep-promoting circuits then leads to increased sleep drive, which allows flies to reach the required amount of sleep. This process is known as sleep homeostasis. A "price" of sleep homeostasis is that sustained activation sleep-promoting circuits leads to increased "sleepiness" and increased seizure activity. On the other hand, if the activity of sleep-promoting circuits is decreased (e.g., experimentally with GtACR1 activation or pharmacologically with 5HT1A agonism), seizures are less likely to occur.

that this increased activity is not isolated to sleep-promoting circuits and occurs more broadly throughout the brain (Fig. 5b). Flies awake for 5–8 h exhibit increased intraneuronal calcium concentrations with increased reactivity to external stimuli, possibly driven by an increase size and number of synapses[6,45,46]. Similar widespread increases in cortical excitability are known to occur in rodent models[47] and people[48–51]. We propose that these broad increases in neuronal excitability across the brain drive elevated seizure susceptibility. This is supported by mammalian models demonstrating that sleep restriction increases seizure risk and severity[52–54]. Taken together, our data are consistent with a model wherein activity of the whole brain increases with accumulating sleep pressure.

Our transcriptomic analysis of the dFB after sleep restriction revealed downregulation of 5HT1A. *Drosophila* express five serotonin receptors, including 5HT1A, 5HT1B, 5HT2A, 5HT2B, and 5HT7[55]. The relationship between sleep and serotonin is complex; serotonin binds to multiple receptors in multiple cell types across the brain to produce varied effects on sleep[56]. When 5HT1A is globally disrupted in 5HT1A mutant flies, sleep becomes decreased and fragmented[57]. We show here that a selective 5HT1A agonist, 8-OH-DPAT, also decreases sleep, and that this is predominantly mediated by action at sleep-promoting cells (Fig. 7e); this further emphasizes the complex relationships between sleep and serotonin and the context dependence of the specific cell types and receptors involved in serotonergic signaling. These data are consistent with a model in which sleep restriction leads to loss of 5HT1A in sleep promoting centers. Given the known inhibitory effects of 5HT1A[39], loss of 5HT1A-medited inhibition would promote activity of sleep-promoting centers to lead to increased sleep. Consistent with this model, we demonstrate that RNAi-mediated knockdown of 5HT1A in sleep-promoting centers increases sleep (Fig. 7d). Our manipulations of 5HT1A used the 23E10-Gal4 driver, which targets not only the dFB but also specific ventral nerve cord (VNC) cells that were recently shown to be critical for sleep effects[15,16]. Given that we observed changes in 5HT1A in the dFB, we propose that the decrease in 5HT1A acts in conjunction with other sleep loss-induced changes in the dFB— decreased K⁺ leak conductance[8] and increased Rho-GTPase-activating protein crossveinless-c activity[7]— to promote homeostatic sleep rebound. We also find that promoting 5HT1A activity in the sleep-promoting centers not only decreases sleep, presumably due to decreased activity of sleep-promoting centers, but also decreases seizure burden. Therefore, while increased activity of sleep-promoting centers is important for sleep homeostasis, it comes at the cost of promoting seizures if the neural milieu is seizure-prone (Fig. 9).

The use of serotonergic drugs to treat epilepsy has received growing attention since the FDA approved fenfluramine in 2020 to treat seizures associated with Dravet Syndrome. Fenfluramine is thought to act through 5-HT1D- and 5-HT2C-agonism[58] to improve seizure control. We find that use of 5HT1A agonists can be used to suppress seizures that are worsened with sleepiness in *Drosophila* (Fig. 8). We identify an FDA-approved agonist, buspirone, to decrease seizures sensitive to sleepiness. Broad activation of serotonergic neurons with trh-Gal4>TrpA1 led to worsened seizures (Fig. 4d, e, Supplementary Fig. 12a, b), demonstrating that broadly promoting serotonergic tone may not be helpful and further emphasizing the importance of 5HT receptor specificity. Despite the growing number of anti-seizure medications clinically available, about one-third of individuals with epilepsy continue to have seizures refractory to medical therapy. Interventions that improve sleep quality, such as the ketogenic diet, also improve seizure control[59]. Targeting non-conventional mechanisms, including the effects of sleepiness on seizures, may offer improved clinical management.

## Methods
Adult male mated flies were used, 5–10 days post-eclosion. Flies were raised in *Drosophila* incubators with a 12 h light:12 h dark light cycle at 25° and ~60–65% relative humidity. Flies were maintained on a standard cornmeal-molasses diet, including 64.7 g/L cornmeal, 27.1 g/L dry yeast, 8 g/L agar, 61.6 mL/L molasses, 10.2 mL/L 20% tegosept, and 2.5 mL/L propionic acid. A detailed description of all experimental methodology is located in the Supplementary Information.

### Reporting summary
Further information on research design is available in the Nature Portfolio Reporting Summary linked to this article.

## Data availability
Source data are provided with this paper. The transcriptomics dataset of control versus sleep-deprived neurons from the dorsal fan-shaped body data used in this study are available in the NCBI Sequence Read Archive database under accession code PRJNA1135478. Source data are provided with this paper.

## Code availability
Our scripts to detect sleep and seizures from video tracking data, including the "CynthiSeize" script with detailed user instructions, sample data, and sample outputs, is deposited at Zenodo and is publicly available at https://zenodo.org/records/15619740.

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

## Acknowledgements

We thank Kiet Luu for assistance with fly maintenance. We thank members of the laboratory for helpful discussions. We thank Antoneta Karaj for assistance with statistical analyses. Schematics were created with BioRender (BioRender.com). The article is subject to HHMI's Open Access to Publications policy. HHMI lab heads have previously granted a nonexclusive CC BY 4.0 license to the public and a sublicensable license to HHMI in their research articles. Pursuant to those licenses, the author-accepted manuscript of this article can be made freely available under a CC BY 4.0 license immediately upon publication. The content is the sole responsibility of the authors and does not necessarily represent the official views of the National Institutes of Health. We thank the following funding sources: National Institutes of Health grant K08NS131602 (VAC); National Institutes of Health grant R38HL143613. PI: Peter Klein. (VAC); National Institutes of Health grant T32NS007413. PI: Michael Robinson. (VAC); American Academy of Neurology Neuroscience Research Training Scholarship (VAC); CURE Epilepsy Taking Flight Award (VAC); National Institutes of Health grant P50HD105354 (GLG and MEP); National Institutes of Health grant R01NS048471 and R01DK120757 (AS); Howard Hughes Medical Institute (AS).

## Author contributions

Conceptualization: V.A.C., A.S.; Methodology: V.A.C., C.T.H., A.S.; Investigation: V.A.C., C.T.H., F.V.S., Y.L., C.S., H.M.S., S.K., J.S., C.G., Z.Y., G.L.G., M.E.P., A.S.; Visualization: V.A.C.; Funding acquisition: V.A.C., A.S.; Supervision: V.A.C., A.S.; Writing—original draft: V.A.C., A.S.; Writing—review and editing: V.A.C., C.T.H., Y.L., G.L.G., M.E.P., A.S.

## Competing interests

The authors declare no competing interests.
