## [Transparent Peer Review file · Nature Communications]

Sleep drive, not total sleep amount, increases seizure risk

Corresponding Author: Professor Amita Sehgal

This manuscript has been previously reviewed at another journal. This document only contains information relating to versions considered at Nature Communications. Mentions of the other journal have been redacted.

Version 0:

Reviewer comments:

Reviewer #1

(Remarks to the Author)

Cuddapah et al propose that sleep drive, not sleep amount, is responsible for an increase in seizures, using a fly model of bang-induced seizures. This is a revised version of a manuscript that was previously submitted to [REDACTED]. A tracked version of the changes in the manuscript, or a list of modifications in the rebuttal letter were not provided to this reviewer. However, I infer the main changes are as follows:

- 1 – Many figures that used to be in the supplementary material are now being promoted to main figures, making the narrative more accessible.
- 2 – Technical details of the software introduced for seizure analysis are now available.
- 3 – Two experiments were performed to address some comments from Rev 3, but they are not included in the manuscript (they are not key experiments, anyway).

In particular:

- Fig 1 was Extended Fig 1
- Fig 2 was Fig 1
- Fig 3 was Fig 2 - with the addition of p(doze) analysis
- Fig 4 was Fig 3
- Fig 5 was Extended Fig 11
- Fig 6 was Fig 4
- Fig 7 was Fig 5
- Fig 8 was Extended Fig 15

The revision process has slightly improved the narrative and appropriately described one of the main methodologies. However, the experimental work and conclusions remain essentially unchanged.

Initially, I assessed that the conclusions were not supported by the data. Since neither the data nor the conclusions have changed, it remains difficult for me to alter my assessment. The previous manuscript version was not very accessible, a point also noted by other reviewers. Upon revisiting the revised manuscript, I identified some aspects previously overlooked, but these mostly highlight new weaknesses.

In particular,

1) The manuscript is titled "Sleepiness, not total sleep amount, increases seizure risk". Among 23 figures, only the experiment in Figure 3 attempts to dissociate sleepiness from sleep amount. All other experiments relate to reducing sleep amount but do not address sleepiness or sleep drive directly. They manipulate overall sleep amount, for instance, with caffeine or thermogenetics, but fail to differentiate between the effects of sleep amount and sleepiness. This strong claim is based on a single experiment.

2) The experiment in Figure 3, which ostensibly supports this claim, has its own issues. It involves thermogenetic

manipulation of two GAL4 lines previously shown to induce sleep deprivation with and without rebound (c584, Dubowy et al; Tdc2, Seidner et al), with added quantification of sleep rebound using Griffith's p(Doze) method. However, the evidence remains inconclusive. If the authors wish to support their conclusions robustly, they should provide more comprehensive data. They have opted not to explore other methods of sleep interference, but at least a broader range of GAL4 lines could be used. For example, in Dubowy et al., the Seghal lab described two GAL4 lines that led to sleep deprivation with rebound: c584 and 104906; as well as two lines without rebound: MJ63 and C453. Why not use these? The experiments are performed only in males, and thus, by recombining uas-TrpA1 with tko, it would be straightforward to test numerous GAL4 lines. Moreover, Figure 1 clearly shows that the effect of caffeine is much stronger in eas (95 seizures vs. 1) than in tko (10 vs. 1), making the focus on tko alone reductive and incomplete.

3) Figure 4 shows that c584 is the only GAL4 line that, when thermogenetically activated, results in a less severe seizure phenotype: fewer seizures and shorter duration. This contradicts the key finding in Figure 3. The comparison is complicated since the stimulation in Figure 3 lasts 12 hours at 30°C, while in Figure 4 it lasts only 4 minutes at 25°C. The disparity in conditions makes it difficult to draw a conclusion. Additionally, the link between a 4-minute TrpA1 activation and sleep (let alone sleepiness) is unclear.

4) An increase in CalexA signal after sleep restriction has been widely reported in literature and likely correlates with the increase in synaptic strength initially shown by the Cirelli lab. While the experiment in Figure 5 confirms these findings, it does not add substantial value to the narrative linking seizures and sleepiness.

5) The experiments in Figures 6 and 7 manipulate "sleep-promoting circuits" as labeled by the combination of 23E10 and 201y. The number of neurons manipulated here is too high, considering 201y essentially labels the entire mushroom bodies (Aso et al. 2009). This may offer insights into seizure control but does little to elucidate the relationship with sleep.

Overall, I am confused about the strategy adopted in this study. The main conclusion of the paper is not supported by the data: only one experiment indirectly addresses sleep rebound as a proxy for sleepiness, and it is a very limited experiment overall. Suggestions to improve the dataset were offered by reviewers 1 and 4, but dismissed. The authors assert that "the strength of our study is that we use five different methods to restrict sleep and consistently find when sleep need is increased, seizures are worsened," but they must acknowledge that none of these methods directly address sleep need.

(Remarks on code availability)

Reviewer #2

(Remarks to the Author)

I am satisfied with the reviewers response to my comments and do not have any additional responses.

(Remarks on code availability)

Reviewer #3

(Remarks to the Author)

The authors have addressed the concerns that I raised on their initial submission to [REDACTED]. However, I remain somewhat unconvinced that their automated seizure analysis works as they state. As an example, in supp Video 2 – there are many flies which do not move for the entirety and, moreover, are clearly lying on their sides. This suggests that these flies are dead. The resolution is still extremely poor and the magnification of the one well depicting a 'seizure' fly low. Perhaps it might be better to show just a few wells surrounding the seizing fly? Can a normal visible light camera be used (instead of IR), for the purpose of illustrating the software only, to show clearer images at higher mag?

(Remarks on code availability)

Version 1:

Reviewer comments:

Reviewer #1

(Remarks to the Author)

The authors have performed a series of new experiments as suggested that in my opinion helps framing the work and strengthen the conclusions. I still think that a smoking gun is missing but I concede that the authors have addressed the issue from very different angles and that creates a solid enough reference. I am in favour of publication of the paper as it stand experimentally but I would suggesting renaming "sleep need" to "sleep drive". They are not assessing need - which is

a term loaded of significance especially at the cell biological level - but drive.

(Remarks on code availability)

Reviewer #3

(Remarks to the Author)

The clarity of the added video (new Supplementary Video 4) is better and, for the fly highlighted, shows seizure-like activity. However, it is a concern that in the 4 rows of wells shown (A - D) 20 out of 32 flies are dead (i.e. they do not move for the entirety of the recording and have a 'crumpled' posture). If this death rate is typical, then this should be clearly stated in the methods, along with a statement that all wells were manually assessed for death and those flies removed from any analysis. Moreover, given that lethality is over 50%, what does this say about the viability of those flies that are still alive?

(Remarks on code availability)

Code is not provided (but I am no expert)

REVIEWER COMMENTS

Reviewer #1 (Remarks to the Author):

Cuddapah et al propose that sleep drive, not sleep amount, is responsible for an increase in seizures, using a fly model of bang-induced seizures. This is a revised version of a manuscript that was previously submitted to [REDACTED]. A tracked version of the changes in the manuscript, or a list of modifications in the rebuttal letter were not provided to this reviewer. However, I infer the main changes are as follows:

- 1 – Many figures that used to be in the supplementary material are now being promoted to main figures, making the narrative more accessible.
- 2 – Technical details of the software introduced for seizure analysis are now available.
- 3 – Two experiments were performed to address some comments from Rev 3, but they are not included in the manuscript (they are not key experiments, anyway).

In particular:

- Fig 1 was Extended Fig 1
- Fig 2 was Fig 1
- Fig 3 was Fig 2 - with the addition of p(doze) analysis
- Fig 4 was Fig 3
- Fig 5 was Extended Fig 11
- Fig 6 was Fig 4
- Fig 7 was Fig 5
- Fig 8 was Extended Fig 15

The revision process has slightly improved the narrative and appropriately described one of the main methodologies. However, the experimental work and conclusions remain essentially unchanged.

Initially, I assessed that the conclusions were not supported by the data. Since neither the data nor the conclusions have changed, it remains difficult for me to alter my assessment. The previous manuscript version was not very accessible, a point also noted by other reviewers. Upon revisiting the revised manuscript, I identified some aspects previously overlooked, but these mostly highlight new weaknesses.

In particular,

- 1) The manuscript is titled "Sleepiness, not total sleep amount, increases seizure risk". Among 23 figures, only the experiment in Figure 3 attempts to dissociate sleepiness from sleep amount. All other experiments relate to reducing sleep amount but do not address sleepiness or sleep drive directly. They manipulate overall sleep amount, for instance, with caffeine or thermogenetics, but fail to differentiate between the effects of sleep amount and sleepiness. This strong claim is based on a single experiment.

Our finding that sleepiness increases seizure risk is not based on results of a single experiment.

The Miesenbock lab showed that sleep need is encoded by increased activity of the dorsal fan-shaped body neurons (PMID: 28898631, 30894743). In addition, the Nitabach lab identified mushroom body neurons that encode sleep need (PMID: 26455303).

We used thermogenetics as well as two optogenetic tools to find that hyperactivation of these neurons encoding sleep need increases seizure severity. Activation of dorsal fan shaped body neurons or mushroom body neurons each separately worsen seizure severity (Fig. 4), even when total sleep amount is unchanged. Sustained co-activation of these sleep need-encoding neurons with *csChrimson* worsens seizures (Ext. Data Fig. 14). Perhaps most importantly, inhibition of sleep-need encoding neurons with *GtACR1* protects against seizures, even when flies have slept less (Fig. 6).

These experiments reveal that seizure severity tracks closely with sleep need, and not sleep amount.

To further test our findings, we now perform the experiments recommended by Reviewer #1, which further substantiate our conclusions. As previously recommended, we starved bang-sensitive flies for 12 hours and tested seizure severity. The Shaw lab has demonstrated that 12 hours of starvation leads to sleep loss without rebound sleep (PMID: 20824166). We find that starvation does not lead to worsening of convulsive seizures (Ext. Data Fig. 11 and also below). There were subtle prolongations of the paralysis time from 23.1 to 24.8 seconds and recovery time from 6.6 to 9.0 seconds, but these were thought to be relatively minor changes. This further substantiates our finding that only when sleep need increases, and not just sleep loss, does seizure severity increase.

Ext. Data Fig. 11 | Sleep restriction after starvation does not increase convulsive time or induced seizure time in *tko^{25t}* mutant flies. a-e, *tko^{25t}* flies were starved for 12 hours then seizures were induced. There was no significant prolongation of convulsive (tonic/clonic) or total seizure time as compared to fed flies. Starvation is associated with sleep loss without sleep rebound³³. n = 48-56 flies/condition. Unpaired two-tailed t-test was used. *p<0.05. Data are presented as mean values ± SEM.

In addition, as recommended, we now deprive flies of sleep using two additional Gal4 drivers from the referenced Dubowy et al study (PMID: 26951392) (please also see Reviewer #1 Point #2 below). We drove expression of *TrpA1* in 2 additional wake-promoting subsets of neurons using the *c453-Gal4* and *104906-Gal4* drivers. Sleep loss induced by *c453-Gal4>TrpA1* activation is associated with no sleep rebound, while sleep loss induced by *104906-Gal4>TrpA1*

activation is associated with rebound sleep (PMID: 26951392). As presented in Ext. Data Fig 10 (and below), we find that only 104906-Gal4>TrpA1 activation, which is associated with increased sleep need, increases seizure severity as exhibited by a prolonged convulsive time (Ext. Data Fig. 10C) and total seizure time (Ext. Data Fig. 10D). The extension in convulsive time appears to correlate with a reduced initial paralysis phase (Ext. Data Fig. 10B) even though total seizure time was extended (Ext. Data Fig. 10D).

We find that these experiments suggested by the Reviewer further substantiate our conclusion that sleep need increases seizure severity.

2) The experiment in Figure 3, which ostensibly supports this claim, has its own issues. It involves thermogenetic manipulation of two GAL4 lines previously shown to induce sleep deprivation with and without rebound (c584, Dubowy et al; Tdc2, Seidner et al), with added quantification of sleep rebound using Griffith's p(Doze) method. However, the evidence remains inconclusive. If the authors wish to support their conclusions robustly, they should provide more comprehensive data. They have opted not to explore other methods of sleep interference, but at least a broader range of GAL4 lines could be used. For example, in Dubowy et al., the Seghal lab described two GAL4 lines that led to sleep deprivation with rebound: c584 and 104906; as well as two lines without rebound: MJ63 and C453. Why not use these? The experiments are performed only in males, and thus, by recombining uas-TrpA1 with tko, it would be straightforward to test numerous GAL4 lines. Moreover, Figure 1 clearly shows that the effect of caffeine is much stronger in eas (95 seizures vs. 1) than in tko (10 vs. 1), making the focus on tko alone reductive and incomplete.

We now include the additional experiments suggested by the Reviewer and find that our conclusions are further supported (please also see Reviewer #1 Point #1 above). We now include the following in the Results section of the manuscript:

“To further extend these findings, we activated two additional subsets of neurons labeled by the c453-Gal4 and 104906-Gal4 drivers that have previously been demonstrated to lead to sleep

loss³⁰. TrpA1-mediated activation of neurons expressing c453-Gal4, which leads to sleep loss without sleep rebound³⁰, does not worsen seizures (Ext. Data Fig. 10). In contrast, activation of TrpA1 driven by 104906-Gal4, which leads to sleep loss followed by sleep rebound³⁰, exacerbates seizures (Ext. Data Fig. 10). To test a more naturalistic stimulus of sleep loss that does not lead to sleep rebound³³, we starved flies for 12 hours and found that seizures are not significantly worsened (Ext. Data Fig. 11). In conjunction with the effects of sleep need on spontaneous seizures, these experiments implementing thermogenetic- and starvation-induced sleep loss consistently indicate that manipulations increasing sleepiness, but not sleep amount per se, drive seizure severity.”

We’d like to again emphasize that we believe a strength of the current study is that we use multiple complementary methods of sleep interference. Every method of sleep interference has pros and cons. In total we use 8 methods to deprive sleep and find that only when sleep need is increased does seizure severity worsen. This is detailed in a revised Discussion:

“We use eight complementary methods to restrict sleep [(1) caffeine, (2) two short-sleeping mutants, thermogenetic wake-promoting neuron activation with (3) c584>TrpA1, (4) Tdc2>TrpA1, (5) c453>TrpA1, and (6) 104906>TrpA1, (7) starvation, and (8) optogenetic sleep-promoting neuron inhibition with 23E10, 201y>GtACR1] and consistently find when sleep need is increased, seizures are worsened.”

We do not focus on *tko*^{25t} mutant flies alone; Figures 6, 7, and 8 further test our initial findings from *tko*^{25t} mutant flies using a pharmacological model for seizures- picrotoxin. In total, we show that sleep loss in flies leads to worsened seizures using 2 neurogenetic fly models of epilepsy and 1 pharmacological model of epilepsy.

3) Figure 4 shows that c584 is the only GAL4 line that, when thermogenetically activated, results in a less severe seizure phenotype: fewer seizures and shorter duration. This contradicts the key finding in Figure 3. The comparison is complicated since the stimulation in Figure 3 lasts 12 hours at 30°C, while in Figure 4 it lasts only 4 minutes at 25°C. The disparity in conditions makes it difficult to draw a conclusion. Additionally, the link between a 4-minute TrpA1 activation and sleep (let alone sleepiness) is unclear.

There is no contradiction. A key difference between the experiments in Figures 3 and 4 is whether or not total sleep amount is manipulated.

In Figure 4, the goal was to understand how subpopulations of neurons affect seizure severity independent of changes in sleep. Therefore, we used a 4-minute activation to not change total sleep amount. Given that sleep is here defined as immobility for at least 5 minutes, this 4-minute activation would not change total sleep. We now make this point clearer in the main text. c584-Gal4>TrpA1 is activated for 4 minutes at 25°C to understand the effects of acute activation of c584 neurons on seizure severity (see Figure 4a). As stated on lines 208-210, “This activation temperature was selected because it adequately increases TrpA1 conductance³³ and also avoids direct effects of temperature on bang-sensitive seizures in the *tko*^{25t} background²⁹.” We demonstrate that acute activation of c584 neurons does not worsen seizures (and actually has a protective effect).

In contrast, total sleep amount is reduced in Figure 3. To accomplish this, c584 neurons are activated (c584-Gal4>TrpA1) for 12-24 hours, and then effects on seizure severity are

assessed. To avoid the direct effects of 30°C on seizure severity, “flies were brought to room temperature for 20 minutes prior to seizure testing” (see lines 181). Here when c584 are chronically activated, causing decreased sleep and subsequent sleep rebound, seizures are worsened.

These experiments allowed us to disambiguate direct effects of c584 neurons on seizure severity from indirect effects through sleep behavior. This led us to the conclusion that sleep loss leads to worsened seizures when there is high sleep need.

4) An increase in CaLexA signal after sleep restriction has been widely reported in literature and likely correlates with the increase in synaptic strength initially shown by the Cirelli lab. While the experiment in Figure 5 confirms these findings, it does not add substantial value to the narrative linking seizures and sleepiness.

The Reviewer asserts that increases in CaLexA signal after sleep restriction have been widely reported in the literature; we ask the Reviewer to kindly provide references for this statement. As we report, “Consistent with previous findings, we found increased intracellular calcium in sleep-promoting brain regions including the dFB and EB^{7,9}, as well as the MB (Fig. 5a, c-e)” (lines 241-242). Increases in intracellular calcium have been observed in sleep-promoting brain regions, but the brain-wide increases we report here are novel. We do not know whether these correlate with increases in synaptic strength.

The finding that broad, brain-wide increases in intracellular calcium occur after sleep restriction is highly relevant for seizures. Seizures are manifestations of hypersynchronous hyperactivation of neuronal circuits, and this correlates with increases in intracellular calcium.

5) The experiments in Figures 6 and 7 manipulate "sleep-promoting circuits" as labeled by the combination of 23E10 and 201y. The number of neurons manipulated here is too high, considering 201y essentially labels the entire mushroom bodies (Aso et al. 2009). This may offer insights into seizure control but does little to elucidate the relationship with sleep.

We did see in our Gal4 screen (see Figure 4) that separate activation of the dorsal fan-shaped body and mushroom body each independently worsened seizure severity. We coupled them in the manipulations done in Figures 6 and 7 to get the most robust effects for dissecting mechanism. As the Reviewer indicates, there are likely subpopulations of neurons in these regions that are responsible for the increase in seizure risk after sleep loss. We plan to identify these in the future.

The number of neurons manipulated does not affect our major finding, which is that activity of neurons that encode sleep need are responsible for the increase in seizure risk after sleep loss. We identify the sleep-promoting circuits that drive this association. Using transcriptomics, we then identify a serotonergic mechanism by which sleep-promoting neurons increase in activity after sleep loss and implicate this as a target for seizure control.

Overall, I am confused about the strategy adopted in this study. The main conclusion of the paper is not supported by the data: only one experiment indirectly addresses sleep rebound as a proxy for sleepiness, and it is a very limited experiment overall. Suggestions to improve the dataset were offered by reviewers 1 and 4, but dismissed. The authors assert that "the strength of our study is that we use five different methods to restrict sleep and consistently find when sleep need is increased, seizures are worsened," but they must acknowledge that none of these

methods directly address sleep need.

Again, we point out that we directly manipulate structures known to encode sleep need to directly understand their role in seizure risk, and also use multiple drivers that alter sleep with/without changes in sleep need (see Figures 4, 6, and Ext. Data Fig. 14). Our conclusions are not based upon the results of a single experiment.

As detailed above, we have now performed the additional experiments suggested by the Reviewer and continue to find that that sleep loss associated with sleep need increases seizure severity.

Reviewer #2 (Remarks to the Author):

I am satisfied with the reviewers response to my comments and do not have any additional responses.

We thank Reviewer #2 for their insights and are pleased that we have addressed their comments.

Reviewer #3 (Remarks to the Author):

The authors have addressed the concerns that I raised on their initial submission to [REDACTED]. However, I remain somewhat unconvinced that their automated seizure analysis works as they state. As an example, in supp Video 2 – there are many flies which do not move for the entirety and, moreover, are clearly lying on their sides. This suggests that these flies are dead. The resolution is still extremely poor and the magnification of the one well depicting a 'seizure' fly low. Perhaps it might be better to show just a few wells surrounding the seizing fly? Can a normal visible light camera be used (instead of IR), for the purpose of illustrating the soft ware only, to show clearer images at higher mag?

We thank Reviewer #3 for their previous comments. We believe the manuscript was strengthened by addressing those comments.

Indeed, some flies in the Supplementary Video 2/Supplementary Video 3 were dead. In other experiments, we found that severe, prolonged seizures led to fly death, and provided quantification of this (e.g. Ext. Data Fig. 7d, h, Ext. Data Fig. 8).

As suggested by the Reviewer, in a new Supplementary Video 4, we now show what a representative seizure looks like at higher magnification with visible light. We hope the Reviewer agrees that this higher-resolution video allows for improved visualization of a tonic-clonic seizure, including clonic movements of the wings and legs.

REVIEWERS' COMMENTS

Reviewer #1 (Remarks to the Author):

The authors have performed a series of new experiments as suggested that in my opinion helps framing the work and strengthen the conclusions. I still think that a smoking gun is missing but I concede that the authors have addressed the issue from very different angles and that creates a solid enough reference. I am in favour of publication of the paper as it stand experimentally but I would suggesting renaming "sleep need" to "sleep drive". They are not assessing need - which is a term loaded of significance especially at the cell biological level - but drive.

We now replace the term “sleep need” with “sleep drive”.

Reviewer #3 (Remarks to the Author):

The clarity of the added video (new Supplementary Video 4) is better and, for the fly highlighted, shows seizure-like activity. However, it is a concern that in the 4 rows of wells shown (A - D) 20 out of 32 flies are dead (i.e. they do not move for the entirety of the recording and have a 'crumpled' posture). If this death rate is typical, then this should be clearly stated in the methods, along with a statement that all wells were manually assessed for death and those flies removed from any analysis. Moreover, given that lethality is over 50%, what does this say about the viability of those flies that are still alive?

To generate this video, we did not use our typical video tracking pipeline, whereby flies are recorded for 4 days for seizures using an IR camera. Instead, after the conclusion of a 4-day recording, we video recorded flies using a camera without an IR filter to improve resolution. Given that this experiment could no longer be paired with our automated video tracking software, seizures had to be manually identified by an experimenter, and this was done after the completion of a typical experiment, as seizures are more frequent then.

In the legend for Supplementary Video 4, we now state “This video was taken after the conclusion of typical experimental recording period and observed by the experimenter without using video tracking software.” and “Lethality was noted after picrotoxin-induced seizures and found to be associated with sleep (Ext. Data Fig. 8).”

In the Methods we stated “For hypothesis testing of spontaneous seizure frequency, given the categorical variable (seizure count per day per fly), right-skewed distribution of this dataset, and censoring due to fly death during the experiment, a negative binomial model with Wald test was implemented.” We now further emphasize that dead flies are accounted for in our seizure analyses by stating “This analysis accounted for lethality observed after picrotoxin-induced seizures.”

We agree that a high seizure burden likely affects viability, as also observed in rodent models and people with epilepsy.

Reviewer #3 (Remarks on code availability):
Code is not provided (but I am no expert)

All code is provided at

https://github.com/cthsu86/CynthiSeize/releases/tag/CynthiSeize_v1.1

Code was not provided as part of the manuscript files.